# Developing the Physical Fitness of Children: A Systematic Scoping Review of Pedagogy in Research

**DOI:** 10.3390/sports13090309

**Published:** 2025-09-08

**Authors:** Mark Helme, Ian Cowburn, Kevin Till

**Affiliations:** Centre for Child and Adolescent Physical Literacy, Carnegie School of Sport, Leeds Beckett University, Leeds LS6 3QW, UK; mark.helme@leedsbeckett.ac.uk (M.H.);

**Keywords:** strength, aerobic, power, exercise, health, conditioning, motivational climate, coaching, teaching

## Abstract

Despite a robust body of evidence supporting both the need for and the effectiveness of physical fitness interventions in children aged 5–11, global fitness levels in this age group continue to decline. This systematic scoping review interrogates a critical, often overlooked dimension of this paradox: the pedagogy of fitness-intervention design and delivery. By analysing 106 primary research studies, the review exposes a consistent pattern. Interventions are predominantly highly structured (89%), rarely foster a mastery-oriented motivational climate (only 11%), and fail to report practitioner behaviours (65%). While most interventions yielded positive fitness outcomes, these gains were achieved without the use of pedagogical strategies known to support engagement, autonomy, and long-term adherence in children. This suggests that current approaches may achieve short-term physiological improvements but are limited in cultivating the motivational and developmental conditions necessary for sustained impact. The findings underscore a pressing need for future research to move beyond the “what” of fitness programming and rigorously address the “how.” Embedding and explicitly reporting pedagogical elements—such as supportive practitioner behaviours, autonomy-supportive structures, and mastery climates—could transform fitness interventions into developmentally appropriate, engaging, and sustainable experiences for children. Without this shift, we risk perpetuating interventions that are effective in the lab but ineffective in life.

## 1. Introduction

Globally, concerns about the current and future health of children are escalating, due to declining physical activity levels, movement competence, and fitness, contributing to rising obesity rates [1,2,3,4,5]. Faigenbaum et al. [6] conceptualized this issue as the Paediatric Inactivity Triad (PIT), comprising exercise deficit disorder (i.e., reduced physical activity levels below recommended levels), paediatric dynapenia (i.e., low levels of strength, not caused by illness), and physical illiteracy (i.e., low levels of confidence, competence and motivation to engage in physical activity). These factors contribute to a negative spiral [7], where reduced activity leads to diminished fitness, motor competence, and confidence, creating a proficiency barrier [8] to health-promoting activities, resulting in increased sedentary behaviours.

The World Health Organization (WHO) recommends at least 60 min of moderate-to-vigorous physical activity daily for children aged 5–17 years, including muscle and bone-strengthening activities [9]. Furthermore, cardiorespiratory and muscular fitness are linked to a range of broader potential benefits in children, such as cognitive function and academic achievement [7,9,10], highlighting the importance of fitness development in children. There is also consistent evidence, demonstrated through meta-analysis, for the development of fitness following various training modalities (e.g., resistance [11], plyometric [12]) in children. Despite these evidence-based physical activity guidelines and efficacy of training, fitness levels continue to decline. Faigenbaum and colleagues [13] emphasized that the delivery environment is as critical as the exercise prescription itself. Therefore, exploring fitness interventions in children beyond ‘what’ is delivered to ‘how’ and ‘why’ children engage—or fail to engage—in these activities, is warranted.

In children’s sport and exercise, the way that activities are delivered is often referred to as sport pedagogy. Armour [14] defined sport pedagogy as encompassing knowledge in context, learners and learning, and coaches and coaching (to include teachers and other roles, the term practitioner will be used, moving forward). In essence, it refers to the knowledge practitioners need to help participants learn and develop. There are a wide range of pedagogical approaches available for coaches, including non-linear pedagogy [15], teaching games for understanding [16] and blocked practice [17]. An overview of these typologies for coaching games has been provided by Price et al. [18]. The Coaching Practice Planning and Reflective Framework (CPPRF; [19]) is a thinking tool to help practitioners consider how to apply the various pedagogical approaches. The CPPRF highlights the importance of how activities are designed (activity structure) and how practitioners behave to engage participants and support their development (practitioner behaviour). This is especially important in children’s fitness, where both physical development and sustained engagement are key goals. As children typically have a low training age, fitness interventions should be seen more as learning experiences than as programs focused solely on physiological adaptation. Therefore, activities should be designed with a pedagogical approach, rather than one based purely on biomechanics or physiology.

The goal of the CPPRF [19] is constructive alignment between all four elements. While we can, and sometimes do, look at the components separately (e.g., activity design or coach behaviours), the reality is that the activities and behaviours work together to achieve the desired improvement in participants over a period of time and to engage them effectively in the session. Thus, in addition to considering activities and behaviours in isolation, we might consider how they work together to engage participants, for example, and here we might consider theories of motivation. One motivational theory is Self-Determination Theory (SDT; [20]), whereby autonomy, competence, and relatedness foster self-determined motivation, which has been validated in physical education (PE) and youth sport contexts [21,22]. To satisfy these psychological needs in children, practitioners may employ low-structured and co-operative activities, social interaction and autonomy supportive behaviours as pedagogical strategies to increase the motivational climate [18]. Furthermore, Achievement Goal Theory (AGT; [23]) helps inform the motivational climate as task-oriented individuals seek personal mastery (i.e., improving oneself), while ego-oriented individuals seek superiority (i.e., being better than others). Therefore, practitioners can use pedagogical principles throughout their activity design and delivery, using behaviours which create climates that emphasize motivation, effort and cooperation, and, as such, enhancing participant engagement, as advocated by Faigenbaum and McFarland [13].

In summary, whilst a plethora of research has explored the fitness development of children, to date, limited research has considered the pedagogical principles that may be effective for intervention design and delivery. As such, this paper aimed to review fitness-intervention studies used within children aged 5–11 years from a pedagogical perspective, using a systematic scoping review. A scoping review was deemed appropriate for several reasons, as the aim was to explore how pedagogy is reported and integrated within fitness interventions, rather than to assess the effectiveness of specific training modalities. This aligns with the purpose of a scoping review, which is to examine the extent, range, and nature of research activity in a given field [24,25]. Additionally, the flexibility of a scoping review framework allows for the inclusion of a wide range of primary intervention studies with varying methodologies, contexts, and outcome measures, without the constraints of a narrowly defined systematic review [24,25].

## 2. Method

### 2.1. Design and Search Strategy

A systematic scoping review was conducted in line with the Preferred Reporting Items for Systematic Reviews and Meta-Analyses (PRISMA) [26] and the PRISMA extension for scoping reviews guidelines [27]. The systematic scoping review checklist is included as Appendix A.

### 2.2. Identification

A literature search for original articles was undertaken using SPORTdiscus, Medline, and Academic Search Complete databases, between 1st January 2012 and 30th December 2023. An iterative data mining and sampling approach was used to construct a search phrase from key words, to refine the search outputs to relevant sources. The search strategy was the following:

Fitness OR “Motor competence” OR “Motor development” OR “Motor ability” OR “Motor performance” OR “Motor skill” OR “Physical literacy” OR “Fundamental movement skills” OR “Long-term athlete development” OR Athlet* OR injury OR Power OR plyometric* OR Strength OR “Resistance Training” OR Sprint OR Speed OR Endurance OR Aerobic OR Anaerobic OR Conditioning OR Training OR exercise

AND

Youth OR Child*

NOT

disorder* OR abnormal* OR disab* OR deficit* OR “Cerebal palsy” OR “video games” OR syndrome* OR Patien* OR Kidney OR Liver OR disease

The inclusion/exclusion criteria are displayed in Table 1, and were used to conduct a standardisation process on 29 randomly selected papers. Reviewers either rejected or accepted studies for further review based initially on their title, followed by abstract and, finally, reading of the full text. There was 100% agreement between reviewers with 20 titles rejected on title, four rejected on abstract, no papers rejected after reading the full text, and five accepted for review, from the sample of 26. Therefore, the inclusion and exclusion criteria, displayed in Table 1, were accepted and applied to the remaining titles from the search. On completion of screening, the reference lists from all accepted papers were then screened for any relevant studies not found through the original search. These papers were then screened from their abstract and then full text, before inclusion.

### 2.3. Inclusion Criteria

Peer-reviewed primary data-only studies were included where the mean age of the participants was between 5 and 11 years, reflecting the ages of primary school children, (based on the UK educational system). This age range also reflected early and mid-childhood, prior to the onset of puberty/adolescence. These age ranges were selected as they represent the earliest experiences of a participant and are therefore formative in a child’s perception of fitness training. Additionally, children of this age have different physiological responses to exercise compared to adults [28], necessitating a differentiated prescription of fitness training. The participants in the studies were considered healthy and free from injury, disease or impairment (sensory, cognitive or physical). The intervention studies included at least a component of focus on increasing one or more fitness qualities in either the experimental or control group. Research must have been published in or after 2012. This date reflects a period (12 years) of contemporary literature. The date of 2012 was specifically chosen as the year of publication of the youth physical development model [29], which was deemed be significant in the advancement of fitness training in children.

### 2.4. Exclusion Criteria

Search results were excluded where the participants were not within the specified age ranges, which are presented in Table 1; for example, interventions conducted in pre-schools or high schools. The interventions were sport-specific, or lacked a direct intention to develop at least one fitness quality. Studies that used specific clinical populations, such as those with metabolic conditions, specific impairments or post-injury were excluded. Any studies that were not primary data collection interventions, such as systematic reviews and meta-analyses, were also excluded.

### 2.5. Data Charting

Following the guidelines provided by both Tricco et al. [27] and Arksey and O’Malley [24], the data was charted independently by the lead author (MH). A sample of these studies were reviewed the co-authors (KT and IC). There were no disagreements between reviewers, and the lead author’s charting of the data was agreed to be a valid analysis of the studies sourced. The lead author (MH) extracted the data using a specifically designed Excel spreadsheet. This included descriptive data of participant demographics, intervention, duration, intervention context, intervention leaders and the outcome measures. Participants’ ages were organised into one-year intervals, based on the mean age of the children reported in each study. The duration of all the interventions were reported in the number of weeks; where studies only reported the duration in months, these were standardised to 4.5 weeks per month, for analysis. The context of each study was defined as the nature of the providing organisation and refined into four possible options: schools, community sports clubs, elite sports clubs, and recruited research sample. Post hoc analysis of the studies indicated the following categories of roles who delivered the interventions: coaches, teachers, research leads, research assistants, S&C coaches, instructors, and qualified specialists. Similarly, the nature of the interventions was identified as the following:

A replacement whole session (“Whole session”) from an existing provision, such as one PE class in a school.

A warm-up intervention replacing the initial segment, typically 10 to 20 min in duration, of their existing curriculum delivery (“Warm-up”).

Additional content to an existing provision, such as a voluntary after-school activity. (“Additional content”).

To appraise the pedagogy, the methods of all studies included in this review were analysed for activity structure, planned practitioner behaviours and overall motivational climate. Muir et al. [19] describes pedagogy as an integration and alignment of its component parts (activity structure, practitioner behaviours, and participant engagement) to form a coherent strategy. However, to present the extracted data from the studies in the most accessible form possible, each of the facets were examined separately. To differentiate between the pedagogical concepts, practitioner behaviours were considered those concerned with the interaction between the practitioner and the participants [30]. Therefore, activity structures in this review were defined as those facets of pedagogy which were not specifically practitioner–athlete interactions and were more related to the design of the session activities.

### 2.6. Practitioner Behaviours

Practitioner behaviour analysis was derived from the several assessments which included the coach behaviour assessment system [31], the Arizona State University observation instrument [32], the coach analysis and intervention system [33] and the assessment of coaching tone [34]. As the interventions were conducted in a variety of contexts, such as teachers delivering sessions in schools, the coaching-based tools may not have captured all the reported behaviours. A post hoc iterative approach was adopted, to include any behaviour by the supervising adult that was not included in the aforementioned coach behaviour assessment tools. A binary recording system (1 or 0) was employed to report if the identified behaviours were present or absent from the methods of each study. For each study, a total sum of behaviours was calculated to determine the total number of practitioner behaviours deployed.

### 2.7. Activity Structures

The activity structure of the studies was analysed according to the delivery format, implementation of the SDT [20], and degree of adult supervisory control. The intervention formats were analysed using categories created through post hoc analysis of the methods reported. These delivery formats were linear exercise prescription (LEP) (akin to traditional adult resistance training of sets and reps in a defined exercise order), circuit training, interval training, games (individual), games (pairs), games (small-sided), mixed formats, or not specified. Analysis of SDT was achieved through a yes/no approach when identifying statements in each method that explicitly related to competence, autonomy, or relatedness to one of these three constructs. A recording of ‘no’ represents the fact that no statements were made relating to that construct or that there was a statement considered to be antagonistic to that component’s development. Each study format was then classified as being either low-, moderate-, or high-structure, based on the degree of adult supervisory control, as outlined by Barreiro and Howard [35].

### 2.8. Motivational Climate

Using the activity structure and practitioner behaviour analysis, together with any unclassified content from the methods, each study was then judged as being of mastery/competence, ego/performance or ‘unclear’ climate. To be considered a mastery climate, studies were required to articulate an intention to have a mastery-based approach, which permeates both structure and behaviour and goes beyond an attempt to instruct and give feedback on proper exercise technique. The integration of both AGT and SDT it is a long-standing way of observing and developing coaches, providing a strong rationale for this approach in reviewing the motivational climate [34,36,37].

### 2.9. Descriptive Statistics

All data was extracted into Microsoft Excel for analysis. The frequency of study characteristics such as participant demographics, and the number of studies measuring different factors and using different methods, were quantified to reflect the amount of research dedicated to specific areas.

## 3. Results

The literature search initially identified 23,547 records, 9069 of which were duplicates, leaving 14,478 unique records. Following title, abstract, and full-text screening, there were 79 studies which met the eligibility criteria and had full text available (Figure 1). Screening of the reference lists of the included studies yielded a further 57 possible studies. Of these 57 studies, 30 were removed following abstract and full-text reviews. Following the application of the inclusion and exclusion criteria, 106 studies were included for review, as illustrated in Figure 1.

### 3.1. Description of Studies

Table 2 presents a summary of the studies including participants, intervention and outcome measures. Across all 106 studies, 18,321 children were included within the review. These were from 30 countries, spanning six continents, and the global distribution of these participants can be seen in Figure 2. From those studies that reported the sex of participants, 38% were male (*n* = 7047), 31% were female (*n* = 5621), and 31% (*n* = 5653) were from mixed groups that did not specify the split between sexes. The mean sample size was 173 ± 415 children, and sample sizes ranged from 14 to 3895 children. Only two studies [38,39] included children with a mean age of 5 years (*n* = 1484, 8% of participants). The most observed age category was 10-year-olds (*n* = 6717, 37%); however, 11-year-olds were most frequently measured (*n* = 30 studies), but represented only 12% of the total sample population. The effect of one study [40] must be noted, as they conducted a multi-national study across 3895 children, representing 21% of the participants.

Most fitness interventions were undertaken in a school (*n* = 65, 61%) or community sports club (*n* = 23, 22%). From the studies included, 70 (66%) applied interventions with at least one full session per week, 18 (17%) studies involved a warm-up protocol lasting between 10 and 25 min, and a further 18 (17%) provided additional content to an existing programme of activity. The mean duration of the interventions were 18 ± 21 weeks, the shortest period was 4 weeks [41,42,43,44], and the longest studies lasted 2 years [45,46,47,48]. The studies which lasted two years [45,46,47,48] were conducted in schools and an elite sports club. Of the 106 studies reviewed, only 7 [49,50,51,52,53,54,55] used a method that recruited a sample of participants that were specific volunteers for a research intervention project, and the duration of these studies lasted between 4 and 42 weeks.

**Table 2 sports-13-00309-t002:** Descriptive information of the reviewed studies.

Study	Participant Information	Intervention Context	Outcome Measures
Abate Daga et al. [56]	*n* = 40 M = 40 F = 0 Age range (years) = 8–9 Mean age (years) = NS	Country: Italy Context: Community sports club Format: Warm-up Modality: Games (small-sided) Duration (weeks): 12	Lower body Power Standing long jump Change of direction 10 × 5 m Sprint Aerobic fitness Mini Cooper test Sport Specific Shuttle dribble test
Alberty and ČIllÍK [46]	*n* = 40 M = 20 F = 20 Age range (years) = 6 to 7 Mean age (years) = NS	Country: Slovakia Context: School Format: Whole session Modality: FMS Duration (weeks): 104	Lower body Power Standing long jump Change of direction 4 × 10 m shuttles Flexibility Sit and reach Coordination Plate tapping test Other Kneeling overhead volleyball throw Jump with max-effort rotation
Alesi et al. [57]	*n* = 44 M = 44 F = NS Mean age (years) = NS Intervention = 8.8 ± 1.1 Control = 9.0 ± 0.9	Country: Italy Context: School Format: Whole session Modality: Sport, specific (soccer) Duration (weeks): 26	Change of direction Agility test (bespoke) Other Forward Digit Span Test Backward Digit Span Tests Corsi Block Test Visual Discrimination Tower of London test
Almeida et al. [12]	*n* = 160 M = NS F = NS Age range (years) = NS Mean age (years) = 7.9 years	Country: Brazil Context: School Format: Whole Session Modality: Plyometrics Duration (weeks): 12	Lower body Power Standing long jump Upper body strength Handgrip Muscular endurance Curl ups Speed 20 m sprint Change of direction Square test Flexibility Sit and reach Motor competence KTK Aerobic fitness 1 mile time trial
Alonso-Aubin et al. [58]	*n* = 78 M = 78 F = 0 Age range (years) = 6 to 11 Mean age (years) = NS	Country: Spain Context: Elite sports club (Rugby) Format: Warm-up Modality: Integrative neuromuscular training Duration (weeks): 8	Lower body Power Standing long jump Upper Body Power Med Ball throw (2 kg) Muscular endurance Abdominal Curl test Change of direction 5 × 10 m repeat sprint test Coordination Plate Tapping test Motor competence Functional movement screen
Alves et al. [59]	*n* = 128 M = 67 F = 61 Age range (years) = 10 to 11 Mean age (years) = 10.91 ± 0.51	Country: Portugal Context: School Format: Whole session Modality: Plyometrics, interval training Duration (weeks): 8	Lower body Power Counter-movement jump Standing log jump Upper Body Power Medicine-ball throw Speed 20 m sprint Aerobic fitness Multi-stage fitness test
Annesi et al. [60]	*n* = 141 M = 78 F = 63 Age range (years) = 9 to 12 Mean age (years) = 10.0 ± 0.9	Country: USA Context: Community sports club Format: Whole session Modality: Youth fit 4 life Duration (weeks): 41	Muscular endurance Press-ups Aerobic fitness 3 min run distance Psychological Exercise Barriers Self-Efficacy Scale for Children Self-regulation for physical activity Overall negative mood Injury Sport Specific Executive functioning Other
Arabatzi et al. [41]	*n* = 36 M = 21 F = 15 Age range (years) = NS Mean age (years) = 9.30 ± 0.54	Country: Greece Context: School Format: Whole session Modality: Plyometrics Duration (weeks): 4	Lower body Strength Isokinetic ankle dorsiflexion
Avetisyan et al. [61]	*n* = 20 M = 20 F = NA Age range (years) Mean age (years) = 11 ± 0.64	Country: Armenia Context: School Format: Additional content Modality: Resistance training Duration (weeks): 26	Lower body Power Standing long jump Muscular endurance 10 s press-up test Change of direction 4 × 10 m shuttles Psychological Session enjoyment
Barboza et al. [62]	*n* = 191 M = NS F = NS Age range (years) Mean age (years) = NS	Country: Netherlands Context: Community sports club Format: Warm-up Modality: Warm-up Hockey Duration (weeks): 40	Injury Rate, severity and burden
Bogdanis et al. [63]	*n* = 40 M = NS F = NS Age range (years) = NS Mean age (years) = NS	Country: Greece Context: Community sports club Format: Additional content Modality: Plyometrics Duration (weeks): 8	Lower body Power Counter-movement jump (unilateral and bilateral) Squat jump Drop jump Standing long jump Change of direction 5 m + 5 m 180° turn 10 m + 10 m 180° turn
Boraczyński et al. [64]	*n* = 67 M = 67 F = 0 Age range (years) = NS Mean age = 11.2 ± 0.7	Country: Poland Context: Elite sports club Format: Whole session Modality: Soccer-specific, interval training Duration (weeks): 27	Lower body Power Standing long jump Upper body Strength Hand grip Muscular endurance Bent-arm hang Sit-ups Change of direction 10 × 5 m shuttle run Stability Flamingo balance test Flexibility Sit and reach test Coordination Plate tapping test Aerobic fitness Cycle ergometer
Boraczyński et al. [65]	*n* = 75 M = 75 F = 0 Age range (years) = 10 to 11 Mean age (years) = NS	Country: Poland Context: Elite sports club Format: Whole session Modality: Soccer-specific and resistance training Duration (weeks): 52	Sport Specific Soccer-specific motor competence test ×5
Bouguezzi et al. [66]	*n* = 26 M = 26 F = 0 Age range (years) = NS Mean age (years) = NS	Country: Tunisia Context: Elite sports club Format: Whole session Modality: Plyometrics Duration (weeks): 8	Lower body Power Counter-movement jump Five pogo jumps Speed 20 m sprint Change of direction Illinois agility test Sport-Specific Maximal kicking distance
Bryant et al. [67]	*n* = 165 M = 77 F = 88 Age range (years) = 8 to 10 Mean age (years) = 8.3 ± 0.4	Country: United Kingdom Context: School Format: Whole session Modality: Fundamental movement skills Duration (weeks): 6	Lower body Power Counter-movement jump Speed 10 m sprint Psychological Perceived physical competence subscale for children.
Casolo et al. [68]	*n* = 100 M = NS F = NS Age range (years) = 7 to 9 Mean age (years) = 7.5 ± 0.5	Country: Italy Context: School Format: Additional content Modality: Small-sided games Duration (weeks): 13.5	Aerobic fitness Six-minute walking test
Cenizo-Benjumea et al. [69]	*n* = 497 M = 271 F = 226 Age range (years) = NS Mean age (years) = NS	Country: Spain Context: School Format: Whole Session Modality: Fundamental movement skills Duration (weeks): 18	Lower body Power Counter-movement jump Standing log jump Change of direction 4 × 10 m shuttle run Motor competence 3JS test
Chang et al. [70]	*n* = 52 M = 24 F = 28 Age range (years) = 10 to 11 Mean age (years) = NS	Country: Taiwan Context: School Format: Warm-up Modality: Core stability Duration (weeks): 6	Muscular endurance Plank Lateral plank Dynamic curl-up Static curl-up Stability Single-legged balance Flexibility Sit and reach test Motor competence Functional movement screen
Chaouachi et al. [53]	*n* = 63 M = 63 F = 0 Age range (years) = 10 to 12 Mean age (years) = 11 ± 1	Country: Tunisia Context: Specific research sample Format: Whole session Modality: Resistance training Duration (weeks): 12	Lower body Strength Isokinetic dynamometry (knee extension) Lower body Power Standing long jump Speed 25 m Sprint Stability Stork stability test
Costa et al. [71]	*n* = 38 M = 17 F = 21 Age range (years) = 9 to 10 Mean age (years) = 9.1	Country: Portugal Context: School Format: Whole session Modality: Multi-component fitness training Duration (weeks): 12	Motor competence The motor competence assessment Aerobic fitness Yo-Yo Intermittent Recovery Level 1 Children’s Test Psychological Enjoyment level
Cunha et al. [55]	*n* = 18 M = 18 F = 0 Age range (years) = 10 to 12 Mean age (years) = NS	Country: Brazil Context: Specific research sample Format: Whole session Modality: Resistance training Duration (weeks): 12	Lower body Strength Isokinetic dynamometry (Knee extension) Upper body strength Isokinetic dynamometry (Elbow flexion) Aerobic fitness Peak VO_2_ (Treadmill running) Other Body Composition (DXA Scan)
Cvejic and Ostojić [72]	*n* = 178 M = NS F = NS Age range (years) = 8 to 9 Mean age (years) = 9.02 ±0.33	Country: Serbia Context: School Format: Whole session Modality: Multi-component fitness training Duration (weeks): 13.5	Muscular endurance Sit-ups Press-ups Flexibility Sit and reach test Shoulder stretch Aerobic fitness Multi-stage fitness test
de Greeff et al. [47]	*n* = 499 M = 226 F = 273 Age range (years) = 7 to 9 Mean age (years) = 8.1 ± 0.7	Country: Netherlands Context: School Format: Additional content Modality: Interval training Duration (weeks): 104	Lower body Power Standing long jump Upper body Strength Handgrip Muscular endurance Sit-ups Change of direction 10 × 5 m Shuttle run Aerobic fitness Multi-stage fitness test Executive functioning Golden Stroop test, Digital span backwards Visual span backwards, Wisconsin card-sorting task
Donahoe-Fillmore and Grant [54]	*n* = 26 M = 12 F = 14 Age range (years) = 10 to 12 Mean age (years) = NS	Country: USA Context: Specific research sample Format: Whole session Modality: Yoga Duration (weeks): 8	Flexibility Sit and reach test 90/90 test Coordination Motor competence Bruininks–Oseretsky test of motor proficiency
Drouzas et al. [73]	*n* = 68 M = 68 F = 0 Age range (years): 8 to 11 Mean age (years) = NS	Country: Greece Context: Elite sports club Format: Whole session Modality: Plyometrics Duration (weeks): 10	Lower body Strength Isometric mid-thigh pull Lower body Power Unilateral (CMJ) Bilateral (CMJ), Unilateral squat jump (SJ) Bilateral squat jump (SJ) Standing long jump (SLJ). Speed 20 m sprint Change of direction *T* test
Duncan et al. [74]	*n* = 94 M = 49 F = 45 Age range (years) = 6 Mean age (years) = NS	Country: United Kingdom Context: School Format: Whole session Modality: Integrative Neuromuscular Training Duration (weeks): 10	Lower body Power Counter-movement jump Standing long jump Upper Body Power Seated medicine-ball throw (1 kg) Speed 10 m Sprint Motor competence Test of gross motor competence−2 Psychological Physical self-efficacy
Duncan et al. [75]	*n* = 140 M = 77 F = 63 Age range (years) 6 to 7 Mean age (years) = 6.4	Country: United Kingdom Context: School Format: Whole session Modality: Integrative Neuromuscular Training Duration (weeks): 10	Lower body Power Counter-movement jump Standing long jump Upper Body Power Seated medicine-ball throw (1 kg) Speed 10 m Sprint Motor competence Test of gross motor competence −2 Psychological Perceived motor competence
Duncan et al. [76]	*n* = 124 M = 67 F = 57 Age range (years) = 6 to 11 Mean age (years) = 8.5 ± 1.9	Country: United Kingdom Context: School Format: Whole session Modality: Shuttle time Duration (weeks): 6	Lower body Power Standing long jump Upper Body Power Seated medicine-ball throw (1 kg) Speed 10 m Sprint Motor competence Test of gross motor competence −2
Eather et al. [77]	*n* = 48 M = 29 F = 19 Age range (years) = 10 to 12 Mean age (years) = 10.9 ± 0.7	Country: Australia Context: School Format: Whole session Modality: Multi-component fitness training Duration (weeks): 8	Upper Body Power Seated Basketball throw Muscular endurance Wall squat Press-ups Sit-ups (×7) Flexibility Sit and reach test Aerobic fitness Multi-stage fitness test Psychological Physical-fitness testing experience and attitudes towards physical-fitness testing questionnaire
Elbe et al. [52]	*n* = 300 M = 142 F = 158 Age range (years) = 8 to 10 Mean age = 9.30 ± 0.35	Country: Denmark Context: Specific research sample Format: Whole session. Modality: Resistance training, Interval training and small-sided games Duration (weeks): 42	Aerobic fitness Yo-Yo Intermittent Recovery Level 1 Children’s Test Psychological Physical activity enjoyment scale Youth Sport environment questionnaire
Faigenbaum et al. [78]	*n* = 41 M = NS F = NS Age range (years) 9 to 10 Mean age = NS	Country: USA Context: School Format: Warm-up Modality: Integrative Neuromuscular Training Duration (weeks): 8	Lower body Power Standing long jump Single-legged hop Muscular endurance Push-up test Curl-up test Change of direction 4 × 10 m shuttle run Stability Single-legged balance Flexibility Sit and reach test Aerobic fitness 0.8 km time trial run
Faigenbaum et al. [79]	*n* = 40 M = 16 F = 24 Age range (years) = 7 Mean age = 7.6 ± 0.3	Country: USA Context: School Format: Warm-up Modality: Integrative Neuromuscular Training Duration (weeks): 8	Lower body Power Standing long jump Single-legged hop Muscular endurance Push-up test Curl-up test Change of direction 4 × 10 m shuttle run Stability Stork balance test Flexibility Sit and reach test Aerobic fitness 0.8 km time trial run.
Fernandes et al. [80]	*n* = 71 M = 71 F = 0 Age range (years) = 8 to 11 Mean age (years) = 9.6 ± 0.7	Country: Portugal Context: School Format: Whole session Modality: Soccer specific Duration (weeks): 45	Lower body Power Counter-movement jump Speed 15 m sprint Aerobic fitness Yo-Yo intermittent endurance test 1
Ferrete et al. [81]	*n* = 24 M = 24 F = 0 Age range (years) = 8 to 9 Mean age (years) = NS	Country: Spain Context: Elite sports club Format: Additional content Modality: Resistance training Duration (weeks): 26	Lower body Power Counter-movement Jump Speed 15 m sprint Flexibility Sit and reach test Aerobic fitness Yo-Yo intermittent endurance test 1
Font-Lladó et al. [82]	*n* = 190 M = 90 F = 100 Age range (years) = 7 to 8 Mean age (years) = 7.43 ± 0.32	Country: Spain Context: School Format: Warm-up Modality: Integrative Neuromuscular Training Duration (weeks): 12	Motor competence Canadian agility and Movement skill assessment (CAMSA)
Gallotta et al. [83]	*n* = 230 M = 130 F = 100 Age range (years) = 8 to 11 Mean age (years) = NS	Country: Italy Context: School Format: Whole session Modality: Circuit training Duration (weeks): 22	Muscular endurance Curl-up test, Push-up test, Trunk-lift test Flexibility Sit and reach test Motor competence Körperkoordinationstest Für Kinder Aerobic fitness Multi-stage fitness test
Hammami et al. [84]	*n* = 20 M = 20 F = 0 Age range (years) = NS Mean age (years) = 11.1 ± 0.8	Country: Tunisia Context: Elite sports club Format: Whole session Modality: Resistance training Duration (weeks): 6	Lower body Strength 1 repetition maximum (back squat). Lower body Power Standing long jump. Three hop test. Speed 30 m Sprint Change of direction Change of direction test Stability Y Balance test
Hernández et al. [85]	*n* = 19 M = 19 F = 0 Age range (years) = NS Mean age (years) = 10.2 ± 1.7	Country: Chile Context: Community sports club Format: Whole session Modality: Plyometrics Duration (weeks): 7	Lower body Power Counter-movement jump Speed 30 m sprint Change of direction *T* test
Homeyer et al. [86]	*n* = 303 M = 162 F = 141 Age range (years) = 7 to 11 Mean age (years) = NS	Country: Germany Context: School Format: Additional content Modality: Fundamental movement skills Duration (weeks): 52	Lower body Power Standing long jump Muscular endurance Sit-up test Press-up test Speed 20 m Sprint Flexibility Sit and reach test Coordination Sideways jumping Balancing backwards Motor competence German Motor Test 6–18
Höner et al. [87]	*n* = 516 M = 234 F = 282 Age range (years) = NS Mean age (years) = 11.90 ± 0.76	Country: Germany Context: School Format: Whole session Modality: Multi-component fitness training Duration (weeks): 8	Lower body Power Standing long jump Speed 20 m Sprint Flexibility Stand and reach test Coordination Sideways jumping Balancing backwards Motor competence German Motor Test 6–18
Jaimes et al. [88]	*n* = 63 M = 63 F = 0 Age range (years) = NS Mean age (years) = 9.2 ± 0.5	Country: Columbia Context: School Format: Whole session Modality: Resistance training Duration (weeks): 8	Lower body Power Abalakov Jump, Counter-movement jump Squat jump Standing long Jump Change of direction 4 × 10 m shuttle run
Jarani et al. [89]	*n* = 760 M = 397 F = 363 Age range (years) = 6 to 10 Mean age (years) = 8.3 ± 1.6	Country: Albania Context: School Format: Whole session Modality: Fundamental movement skills Duration (weeks): 22.5	Lower body Power Standing long jump Change of direction 10 × 5 m shuttle run Flexibility Sit and reach test Coordination Hanging-target throw, Low Jump, Backwards ball throw Motor competence Körperkoordinationstest Für Kinder Aerobic fitness Anderson test
Keiner et al. [45]	*n* = 70 M = 70 F = 0 Age range (years) = 9 to 11 Mean age (years) = NS	Country: Germany Context: Elite sports club Format: Additional content Modality: Resistance training, Plyometrics Duration (weeks): 104	Lower body Power Counter-movement jump Squat jump Drop jump
Ketelhut et al. [90]	*n* = 48 M = 28 F = 20 Age range (years) = 9 to 10 Mean age (years) = 10.7 ± 0.6	Country: Germany Context: School Format: Whole session Modality: Multi-component fitness training Duration (weeks): 13.5	Aerobic fitness Six-minute running test
Koutsandréou et al. [91]	*n* = 71 M = 32 F = 39 Age range (years) = 9 to 10 Mean age (years) = 9.35 ± 0.6	Country: Germany Context: School Format: Whole session Modality: Interval training, Fundamental movement skills Duration (weeks): 10	Motor competence Heidelberg Gross Motor Test Aerobic fitness Multi-stage fitness test Other Letter digit span test
Larsen et al. [92]	*n* = 295 M = NS F = NS Age range (years) = 8 to 10 Mean age (years) = NS	Country: Denmark Context: School Format: Whole session Modality: Circuit training, games (small sided) Duration (weeks): 43	Lower body Power Standing long jump Speed 20 m Sprint Stability Flamingo balance test Coordination A coordination wall
Larsen et al. [93]	*n* = 239 M = NS F = NS Age range (years) = 8 to 10 Mean age (years) = NS	Country: Denmark Context: School Format: Whole session Modality: Interval training, games (small-sided) Duration (weeks): 43	Lower body Power Standing long jump Speed 20 m Sprint Stability Flamingo balance test Coordination A coordination wall Aerobic fitness Yo-Yo Intermittent Recovery Level 1 Children’s Test
Latorre Román et al. [94]	*n* = 114 M = NS F = NS Age range (years) = 8 to 12 Mean age (years) = NS	Country: Spain Context: School Format: Whole session Modality: Small-sided games Duration (weeks): 10	Lower body Power Standing long jump Upper body Strength Hand grip Motor competence Slalom dribble test Aerobic fitness Multi-stage fitness test Executive functioning FIREBRAND. School Aptitude Tests Trail-Making Tests Other Creative Imagination Test for Children
Latorre Román et al. [95]	*n* = 58 M = 48 F = 10 Age range (years) = NS Mean age (years) = 8.72 ± 0.97	Country: Spain Context: Elite sports club Format: Additional content Modality: Contrast training Duration (weeks): 10	Lower body Power Counter-movement jump Standing long jump Squat jump Drop jump Speed 25 m Sprint Change of direction *T* test
Lloyd et al. [42]	*n* = 41 M = 41 F = 0 Age range (years) = 9 Mean age (years) = NS	Country: United Kingdom Context: School Format: Whole session Modality: Plyometrics Duration (weeks): 4	Lower body Power 10 consecutive sub-maximal hops, 5 bilateral vertical hops
Lucertini et al. [96]	*n* = 101 M = 51 F = 50 Age range (years) = NS Mean age (years) = NS	Country: Italy Context: School Format: Whole session Modality: Resistance training Duration (weeks): 26	Lower body Power Abalokov jump Upper body strength Hand Grip strength, Pinch strength Change of direction 10 × 10 m stage shuttle run Stability Single-leg stance stand Flexibility Coordination Plate tapping test Motor competence Harre’s obstacle course Aerobic fitness Multi-stage fitness test
Marta et al. [97]	*n* = 134 M = 63 F = 71 Age range (years) = 10 to 11 Mean age (years) = 10.84 ± 0.47	Country: Portugal Context: School Format: Whole session Modality: Plyometrics, Interval training Duration (weeks): 8	Lower body Power Counter-movement jump Standing long jump. Upper Body Power Medicine-ball throw (1 kg) Speed 20 m sprint. Aerobic fitness Multi-stage fitness test
Marta et al. [98]	*n* = 57 M = 57 F = 0 Age range (years) = 10 to 11 Mean age (years) = NS	Country: Portugal Context: School Format: Whole session Modality: Plyometrics, Suspension training Duration (weeks): 8	Lower body Power Counter-movement jump Standing long jump. Upper Body Power Medicine-ball throw (1 kg) Speed 20 m sprint.
Marta et al. [99]	*n* = 125 M = 58 F = 67 Age range (years) = 10 to 11 Mean age (years) = 10.8 ± 0.4	Country: Portugal Context: School Format: Whole session Modality: Plyometrics, Interval training Duration (weeks): 8	Lower body Power Standing long jump Counter-movement jump Upper Body Power Medicine-ball throw Speed 20 m sprint Aerobic fitness Multi-stage fitness test
Marta et al. [100]	*n* = 118 M = 57 F = 61 Age range (years) = 10 to 11 Mean age (years) = 10.84 ± 0.47	Country: Portugal Context: School Format: Whole session Modality: Plyometrics, Suspension training Duration (weeks):	Lower body Power Counter-movement jump Standing long jump. Upper Body Power Medicine-ball throw (1 kg) Speed 20 m sprint.
Marta et al. [101]	*n* = 125 M = 58 F = 67 Age range (years) = 10 to 11 Mean age (years) = 10.8 ± 0.4 years	Country: Portugal Context: School Format: Whole session Modality: Plyometrics, Interval training Duration (weeks): 8	Lower body Power Counter-movement jump Standing long jump. Upper Body Power Medicine-ball throw (1 kg) Speed 20 m sprint. Aerobic fitness Multi-stage fitness test
Marta et al. [102]	*n* = 125 M = 58 F = 67 Age range (years) = 10 to 11 Mean age (years) NS	Country: Portugal Context: School Format: Whole session Modality: Plyometrics, Multi-component fitness training. Duration (weeks): 8	Lower body Power Counter-movement jump Standing long jump. Upper Body Power Medicine-ball throw (1 kg) Aerobic fitness Multi-stage fitness test
Martinez-Vaicano et al. [103]	*n* = 487 M = 248 F = 239 Age range (years) = 9 to 10 Mean age (years) = NS	Country: Spain Context: School Format: Whole session Modality: Small-sided games Duration (weeks): 36	Lower body Power Standing long jump Upper body Strength Hand grip Muscular endurance Change of direction 4 × 10 m shuttles Flexibility Sit and reach tests Aerobic fitness Multi-stage fitness test
Marzouki et al. [104]	*n* = 137 M = 66 F = 71 Age range (years) = 8 to 11 Mean age (years) = NS	Country: Tunisia Context: School Format: Whole session Modality: Plyometrics Duration (weeks): 4	Lower body Power Standing long jump Squat jump Speed 20 m Sprint Change of direction 5–10-5 (pro-agility) Stability Y balance test Aerobic fitness Multi-stage fitness test
Mayorga-Vega et al. [105]	*n* = 75 M = 34 F = 41 Age range (years) = 10 to 11 Mean age (years) = 11.1 ± 0.4	Country: Spain Context: School Format: Whole session Modality: Circuit training Duration (weeks): 8	Lower body power Standing long jump Muscular endurance Bent-arm hang Sit-ups Aerobic fitness Multi-stage fitness test Psychological Physical Self-Description Questionnaire
Menezes et al. [106]	*n* = 38 M = 38 F = 0 Age range (years) = 6 to 10 Mean age (years) = NS	Country: Brazil Context: Community sports club Format: Warm-up Modality: Integrative Neuromuscular Training Duration (weeks):	Lower body Power Counter-movement jump Speed 20 m sprint Change of direction Change of direction square Flexibility Sit and reach test
MlChailidis et al. [107]	*n* = 45 M = 45 F = 0 Age range (years) = NS Mean age (years) = NS	Country: Greece Context: Community sports club Format: Additional content Modality: Plyometrics Duration (weeks): 12	Lower body Strength 10 repetition maximum (back squat) Lower body Power Counter-movement jump Standing long jump Squat jump Drop jump Five bounds. Speed 30 m sprint Anaerobic fitness 30 s Wingate test Sport Specific Kicking distance Other Testosterone levels
Moeskops et al. [108]	*n* = 34 M = 0 F = 34 Age range (years) = 6 to 11 Mean age (years) = NS	Country: United Kingdom Context: Community sports club Format: Whole session Modality: Integrative Neuromuscular Training Duration (weeks): 8	Lower body Power Drop jump 20 hops Muscular endurance Biering–Sorenson test (trunk) Motor competence Functional movement screen
Moran et al. [109]	*n* = 29 M = 29 F = 0 Age range (years) = NS Mean age (years) = NS	Country: United Kingdom Context: Community sports club Format: Whole session Modality: Resistance training Duration (weeks): 8	Lower body Strength Isometric mid-thigh pull Lower body Power Counter-movement jump Upper body strength Handgrip strength
Ng et al. [110]	*n* = 71 M = 71 F = 0 Age range (years) = 6 to 13 Mean age (years) = 9.82 ± 1.90	Country: Hong Kong Context: School Format: Whole session Modality: Change of direction Duration (weeks): 6	Stability Star balance test Psychological Perceived Physical Ability Scale for Children
Orntoft et al. [111]	*n* = 526 M = 257 F = 269 Age range (years) = 10 to 11 Mean age (years) = 11.1 ± 0.4	Country: Denmark Context: School Format: Whole session Modality: Soccer specific Duration (weeks): 11	Lower body power Standing long jump Stability Flamingo balance test Aerobic fitness Yo-Yo Intermittent Recovery Level 1 Children’s Test
Parsons et al. [112]	*n* = 43 M = 0 F = 43 Age range (years) = 9 to 11 Mean age (years) = 11.1	Country: Canada Context: Community sports club Format: Warm-up Modality: FIFA 11+ Duration (weeks): 16	Lower body Power Counter-movement jump Muscular endurance Plank test Change of direction *T* test Stability Y balance test Motor competence Landing-error scoring system
Pinto-Escalona et al. [113]	*n* = 721 M = 377 F = 344 Age range (years) = 7 to 8 Mean age (years) = 7.4 ± 0.5	Country: Spain, France, Portugal, Germany and Poland Context: School Format: Whole session Modality: Multi-component fitness training Duration (weeks): 52	Stability Y balance test Flexibility Front split test Aerobic fitness Multi-stage fitness test Psychological Strengths and Difficulties Questionnaire Other Academic achievement Physical Activity Questionnaire for Children
Polevoy et al. [114]	*n* = 50 M = 50 F = 0 Age range (years) = 9 to 11 Mean age (years) = NS	Country: Russia Context: School Format: Whole session Modality: Multi-component fitness training Duration (weeks): 10	Lower body power Standing long jump Upper body Strength Handgrip Muscular endurance Squats Change of direction 3 × 10 m Flexibility Sit and reach test
Pomares-Nogueraet et al. [43]	*n* = 23 M = 23 F = 0 Age range (years) = 11 to 12 Mean age (years) = 11.8 ± 0.3	Country: Spain Context: Community sports club Format: Warm-up Modality: FIFA 11+ Duration (weeks): 4	Lower body Power Counter-movement jump Standing long jump Drop jump Speed 20 m sprint Change of direction Illinois agility test Stability Y balance test
Ramirez-Campillo et al. [115]	*n* = 14 M = 14 F = 0 Age range (years) = NS Mean age (years) = NS	Country: Spain Context: Community sports club Format: Additional content Modality: Plyometrics Duration (weeks): 6	Lower body Power Counter-movement jump Standing long jump Drop jump Sport Specific Kicking velocity
Redondo-Tebar et al. [38]	*n* = 1447 M = 748 F = 699 Age range (years) = 4 to 6 Mean age (years) = NS	Country: Spain Context: School Format: Whole session Modality: Small-sided games Duration (weeks): 36	Lower body power Standing long jump Change of direction 4 × 10 m shuttle run Aerobic fitness Multi-stage fitness test
Richard et al. [116]	*n* = 173 M = NS F = NS Age range (years) = 9 to 10 Mean age (years) = 9.56 ± 0.61	Country: USA Context: School Format: Whole session Modality: Circuit training Duration (weeks): 12	Executive functioning Runco Creative Assessment Battery, Bertsch’s test of motor creativity, Exercise self-efficacy, Perception of exercise difficulty
Reyes-Amigo et al. [117]	*n* = 24 M = 16 F = 8 Age range (years) = 8 to 10 Mean age (years) = 10.45 ± 0.90	Country: Chile Context: School Format: Whole session Modality: Multi-component fitness training Duration (weeks): 12	Psychological The International Fitness Scale
Rössler et al. [40]	*n* = 3895 M = NS F = NS Age range (years) = 7 to 12 Mean age (years) = NS	Country: Switzerland, Germany, Czech Republic and Holland Context: Community sports clubs Format: Warm-up Modality: FIFA 11+ Duration (weeks): 52	Injury Injury occurrence, time loss, survival exposure
Rössler et al. [118]	*n* = 122 M = 122 F = 0 Age range (years) = 7 to 12 Mean age (years) = NS	Country: Switzerland, Context: Community sports clubs Format: Warm-up Modality: FIFA 11+ Duration (weeks): 10	Lower body Power Counter-movement jump Standing long jump Drop jump Speed 20 m sprint Sport Specific Slalom dribble, Wall-volley test
Sacchetti et al. [48]	*n* = 497 M = 256 F = 241 Age range (years) = 8 to 9 Mean age (years) = NS	Country: Italy Context: School Format: Additional content Modality: Multi-component fitness training Duration (weeks): 104	Lower body power Standing long jump Upper Body Power Medicine-ball throw Speed Medicine-ball throw Flexibility Sit and reach test Motor competence Forward roll Other Physical Activity Questionnaire for children
Sammoud et al. [119]	*n* = 26 M = 26 F = 0 Age range (years) = NS Mean age (years) = NS	Country: Tunisia Context: Elite sports club Format: Additional content Modality: Plyometrics Duration (weeks): 8	Lower body Power Counter-movement jump Standing long jump Sport Specific Front crawl diving start Front crawl water start with a push-off from the wall Front crawl water start without a push-off from the wall
Savičević et al. [120]	*n* = 128 M = 57 F = 71 Age range (years) = 6 to 7 Mean age (years) = 6.23 ± 0.88	Country: Serbia Context: School Format: Whole session Modality: Multi-component fitness training Duration (weeks): 39	Lower body Power Standing long jump Muscular endurance Hanging pull-ups Speed 20 m sprint Flexibility Sit and reach test Coordination Hand tapping Motor competence Backward polygon Hoop throwing Ball rolling
Schlegel et al. [121]	*n* = 48 M = 25 F = 23 Age range (years) = 10 to 11 Mean age (years) = NS	Country: Czech Republic Context: School Format: Whole session Modality: Street workout Duration (weeks): 6	Muscular endurance Press-ups Sit -ups Hanging hold Plank Flexibility Sit and reach
Sijie et al. [39]	*n* = 37 M = 14 F = 23 Age range (years) = 5 Mean age (years) = NS	Country: China Context: School Format: Whole session Modality: Interval training Duration (weeks): 10	Lower body Power Standing long jump Upper body strength Hand grip test Change of direction 4 × 10 m test Flexibility Sit and reach test
Skoradal et al. [122]	*n* = 392 M = 203 F = 189 Age range (years) = 10 to 12 Mean age (years) = 11.1 ± 0.3	Country: Faroe Islands Context: School Format: Whole session Modality: Small-sided games Duration (weeks): 11	Lower body power Standing long jump Stability Stork balance test Aerobic fitness Yo-Yo Intermittent Recovery Level 1 Children’s Test
St Laurent et al. [51]	*n* = 28 M = 15 F = 13 Age range (years) = 7 to 12 Mean age (years) = 9.3 ± 1.5	Country: USA Context: research specific sample Format: Whole session Modality: Suspension training Duration (weeks): 6	Lower body Power Standing long jump Muscular endurance Trunk Lift 90° Push-Up Modified Pull-Up Change of direction 4 × 10 m shuttle run Motor competence Functional movement screen Other Participation score
Stupar et al. [123]	*n* = 207 M = NS F = NS Age range (years) = 6 to 7 Mean age (years) = NS	Country: Serbia Context: School Format: Whole session Modality: Multi-component fitness training Duration (weeks): 16	Speed 20 m sprint Change of direction Stability Flexibility Coordination Plate tapping test Motor competence Backwards obstacle course Aerobic fitness Psychological Injury Sport-Specific Other
Tatsuo et al. [124]	*n* = 57 M = 33 F = 24 Age range (years) = 7 to 8 Mean age (years) = NS	Country: Japan Context: School Format: Additional content Modality: Agility Duration (weeks): 5	Change of direction Repeated side steps
Thompson et al. [125]	*n* = 51 M = 0 F = 51 Age range (years) = 10 to 12 Mean age (years) = NS	Country: USA Context: Community sports club Format: Warm-up Modality: FIFA 11+ Duration (weeks): 8	Lower body Power Drop jump (unilateral) Drop jump (bilateral) Change of direction 45° cut (planned) 45° cut (unplanned)
Tottori et al. [126]	*n* = 58 M = 33 F = 25 Age range (years) = 8 to 12 Mean age (years) = NS	Country: Japan Context: School Format: Whole session Modality: Interval training Duration (weeks): 4	Lower body Power Standing long jump Muscular endurance Sit-ups Aerobic fitness Multi-stage fitness test Executive functioning Digit span test Tower of Hanoi
Trajković and Bogataj [127]	*n* = 66 M = 0 F = 66 Age range (years) = NS Mean age (years) = 11.05 ± 0.72	Country: Serbia Context: Community sports club Format: Additional content Modality: Integrative Neuromuscular Training Duration (weeks): 10	Lower body Power Counter-movement jump Upper Body Power Med-ball throw Speed 10 m sprint Change of direction *T* test Motor competence Körperkoordinationstest Für Kinder
Trajković et al. [128]	*n* = 36 M = 36 F = 0 Age range (years) = 10 to 12 Mean age (years) = NS	Country: Serbia Context: Community sports club Format: Warm-up Modality: FIFA 11+ Duration (weeks): 8	Lower body Power Standing long jump Speed 20 m sprint Change of direction Illinois agility test Aerobic fitness 30–15 intermittent running test Anaerobic fitness Repeat sprint ability
Trecroci et al. [129]	*n* = 24 M = 24 F = 0 Age range (years) = NS Mean age (years) = 11.3 ± 0.70	Country: Italy Context: Community sports club Format: Warm-up Modality: Jump-rope training Duration (weeks): 8	Stability Y balance test (lower quarter) Motor competence Harre’s circuit test
Tseng et al. [130]	*n* = 55 M = 27 F = 28 Age range (years) = 10 to 12 Mean age (years) = NS	Country: Taiwan Context: School Format: Whole session Modality: FIFA 11 + Kids Duration (weeks): 8	Lower body power Standing long jump Muscular endurance Sit-ups Flexibility Sit and reach test Aerobic fitness 800 m running time trial
Turgutet al. [131]	*n* = 29 M = 0 F = 29 Age range (years) = NS Mean age (years) = NS	Country: Turkey Context: Community sports club Format: additional content Modality: Plyometrics Duration (weeks): 12	Stability Star excursion balance test
Vaczi et al. [132]	*n* = 23 M = 0 F = 23 Age range (years) = NS Mean age (years) = NS	Country: Hungary Context: Elite sports club Format: Additional content Modality: Nordic hamstring exercise Duration (weeks): 20	Lower body Strength Isokinetic dynamometry (knee extension) Lower body Power Counter-movement jump
Vasileva et al. [133]	*n* = 90 M = 44 F = 46 Age range (years) = 7 to 9 Mean age (years) = 7.4 ± 0.3	Country: Spain Context: School Format: Warm-up Modality: Integrative neuromuscular training Duration (weeks): 13.5	Upper body Strength Handgrip Aerobic fitness 800 m running time trial Other Salivary HMW-adiponectin
Vera-Assaoka et al. [134]	*n* = 32 M = 32 F = 0 Age range (years) = NS Mean age (years) = NS	Country: Chile Context: Community sports club Format: Additional content Modality: Plyometrics Duration (weeks): 7	Lower body Strength Five-repetition maximum (back squat) Lower body Power Counter-movement jump Drop jump Five bound test Speed 20 m sprint Change of direction Illinois agility test Aerobic fitness Running time trial (2.4 km) Sport Specific Maximum kicking distance
Wang et al. [135]	*n* = 40 M = 40 F = 0 Age range (years) = 9 to 10 Mean age (years) = NS	Country: China Context: School Format: Whole session Modality: Sport-specific (soccer) Duration (weeks): 10	Lower body Power Standing long jump Upper body strength Hand grip strength Muscular endurance Sit-up, Front bridge, Side bridge Stability Single leg standing (eyes closed) Flexibility Sit and reach test Aerobic fitness Multi-stage fitness test
Waugh et al. [50]	*n* = 20 M = 10 F = 10 Age range (years) = NS Mean age (years) = 8.9± 0.3	Country: United Kingdom Context: Research specific sample Format: Whole session Modality: Resistance training Duration (weeks): 10	Lower body Strength Isokinetic dynamometry (Achilles tendon)
Westblad et al. [136]	*n* = 30 M = 15 F = 15 Age range (years) = NS Mean age (years) = 11.8 ± 0.9	Country: Sweden Context: Community sports club Format: Whole session Modality: Resistance training Duration (weeks): 6	Lower body Power Counter-movement jump Squat jump Speed 30 m sprint
Williams et al. [137]	*n* = 34 M = 17 F = 17 Age range (years) = 11 to 12 Mean age (years) = 11.4 ± 0.67	Country: United Kingdom Context: Community sports club Format: Warm-up Modality: Parkour Duration (weeks): 8	Lower body Power Counter-movement jump Speed 10 m sprint Motor competence Overhead squat
Yanci et al. [138]	*n* = 57 M = 33 F = 24 Age range (years) = NS Mean age (years) = 6.32 ± 0.41	Country: Spain Context: School Format: Whole session Modality: Agility Duration (weeks): 4	Change of direction *T* test
Yanci et al. [44]	*n* = 76 M = 44 F = 32 Age range (years) = NS Mean age (years) = 6.42 ± 0.38	Country: Spain Context: School Format: Whole session Modality: Agility Duration (weeks): 4	Change of direction *T* test
Yapıcı et al. [139]	*n* = 116 M = 116 F = 0 Age range (years) = 7 to 9 Mean age (years) = NS	Country: Turkey Context: School Format: Whole session Modality: Multi-component fitness training Duration (weeks): 12	Lower body Power Counter-movement jump Speed 10 m sprint Stability Flamingo test Flexibility Sit and reach test
Ye et al. [140]	*n* = 261 M = 127 F = 134 Age range (years) = 7 to 9 Mean age (years) = 8.27 ± 0.70	Country: USA Context: School Format: Whole session Modality: Circuit training Duration (weeks): 40.5	Upper body Strength Hand grip Muscular endurance Sit-ups Press-ups Motor competence Kicking speed, Throwing speed, Standing long jump, Hopping Aerobic fitness Multi-stage fitness test
Yildiz et al. [49]	*n* = 28 M = 28 F = 0 Age range (years) = NS Mean age (years) = 9.6 ± 0.7	Country: Turkey Context: Specific research sample Format: Whole session Modality: Resistance training Duration (weeks): 8	Lower body Power 10 m sprint Speed 10 m sprint Change of direction *T* test Stability Y balance test Flexibility Sit and reach test Motor competence Functional movement screen
Zarei et al. [141]	*n* = 31 M = 31 F = 0 Age range (years) = NS Mean age (years) =11.5 ± 0.8	Country: Iran Context: Community sports club Format: Warm-up Modality: FIFA 11+ Duration (weeks): 10	Lower body Strength Isokinetic dynamometry (hip, knee and ankle)
Zhang et al. [142]	*n* = 352 M = 177 F = 175 Age range (years) = 7 to 8 Mean age (years) = 7.8 ± 0.7	Country: China Context: School Format: Whole session Modality: Multi-component fitness training Duration (weeks): 10	Upper body Strength Hand grip Speed 50 m sprint Flexibility 50 m sprint Aerobic fitness Multi-stage fitness test

*n* = number of participants, M = Male, F = Female, NS = not stated.

### 3.2. Intervention Modalities

Studies were grouped according to common forms of fitness modalities, such as resistance training or plyometrics. Where the intervention did not align to a recognised training format or was highly specialised toward a particular exercise modality, they were organised into their own category. A total of 22 different training modalities were used across the studies, including agility (*n* = 3), change of direction (CoD; *n* = 1), circuit training (*n* = 3), contrast training (*n* = 1), FIFA 11+ (*n* = 7), fundamental movement skills (FMSs; *n* = 6), integrative neuromuscular training (INT; *n* = 9), interval training (*n* = 6), jump rope training (*n* = 1), multi-component fitness training (*n* = 5), Nordic hamstring exercise (NHE; *n* = 1), parkour (*n* = 1), plyometrics (*n* = 16), resistance training (*n* = 14), shuttle time (*n* = 1), small-sided games (*n* = 3), soccer specific (*n* = 4), suspension training (*n* = 2), core stability (*n* = 1), exergaming (*n* = 1), street workout (*n* = 1), yoga (*n* = 1), and warming-up hockey (*n* = 1). The studies typically applied only one intervention modality (control conditions excluded); however, seven studies used two different modalities such as plyometrics and interval training [97,102] and one study [52] used three different modalities, including resistance training, interval training and small-sided games.

### 3.3. Outcome Measures

One hundred and thirty-three measures (not including iterations of common tests) were used across the 106 studies to assess the interventions. The most frequently used tests were the standing long jump (*n* = 51), linear sprint (*n* = 38), and counter-movement jump (*n* = 34). The mean number of measures per study was 4.3, with a range from one test up to ten tests [107].

### 3.4. Summary of the Pedagogy of Interventions

The methods of each study were analysed and coded for different facets of pedagogical design using activity structure, practitioner behaviours and motivational climate as the three core themes. The coding denoted if any of the defined variables were present or not in the methods and were reported as frequencies.

### 3.5. Activity Structure

Eight different forms of activity structure organisation were identified, of which the most frequently used was LEP (*n* = 38), followed by a mixed format approach (*n* = 37). Thirteen studies did not provide enough information to determine the format used.

When reviewed for statements relating to SDT, only 13 studies were identified for at least one of the three components, the most frequent being competence (*n* = 12), followed by autonomy (*n* = 9) and then relatedness (*n* = 7). From an intervention structure perspective (defined by the degree of adult supervisory control), 94 studies were identified as highly structured, 11 as moderate, and one as low structure. The study identified as low structure [69] applied a gamified approach in children aged 8–11 years. The 11 studies that were of moderate structure were distributed across the range of ages included in this review (5 to 11 years). High-activity structure studies were applied to participants across all ages. Despite the frequency of resistance training or plyometric interventions within this sample of studies, none of them were of low or medium structure.

### 3.6. Practitioner Behaviours

Table 3 presents a summary of the practitioner behaviours identified in the reviewed studies. Analysis of these studies found that no additional behaviours were stated outside of the initial assessment tool. The mean number of stated practitioner behaviours per study was 0.7 ± 1.2, with 69 studies not reporting any practitioner behaviour. Figure 3 presents the frequency distribution for the number of practitioner behaviours stated in each of the studies. The highest frequency of behaviours reported within a study method was seven [116], which was a moderately structured intervention with 9-year-old children. Figure 3 is a histogram that illustrates the frequency with which intervention research reports prescribed practitioner behaviours. This image indicates a clear skew in the data towards the implementation of minimal practitioner behaviours. From the 34 studies that reported practitioner behaviours, instruction was the most frequently deployed (*n* = 30), followed by corrective feedback, illustrated in Figure 4.

Analysis of practitioner behaviour frequency by intervention modality, across the four most frequently utilised modalities (plyometrics, RT, FMS and INT), indicates that some forms of training modality reported more practitioner behaviours than others. Integrated neuromuscular training had the highest mean practitioner behaviour per study (1.6), followed by plyometrics (1.1), then RT (0.7) and FMS (0.7). By way of comparison, INT studies referred to practitioner behaviours more often than not, with six out of nine studies mentioning at least one practitioner behaviour.

### 3.7. Motivational Climate

Table 4 presents the analysis of the studies, including their motivational climates, showing that 94 studies did not include any statements referring to the environment of their intervention. Of the 12 studies that did, all had statements indicative of a mastery climate, and no studies were found to be of an ego-performance orientation. Within these twelve mastery-focused papers, competence statements were most frequently found (*n* = 10), and only two studies [69,120] included all three elements (competence, relatedness and autonomy).

## 4. Discussion

The aim of this systematic scoping review was to examine the reporting of pedagogy in contemporary child fitness-development research. From the 106 studies included in the review, it is evident that there is a dearth of pedagogical information reported within research studies. Within the 106 studies, 69 reported no practitioner behaviours, 94 did not make any statement relating to SDT, and 94 had insufficient information relating to the motivational climate. Additionally, most studies were of high structure (*n* = 94) and often deployed an LEP approach (*n* = 38); they were practitioner-centred and adult-like in nature. However, many of the interventions reviewed had significantly positive effects on child fitness, compared to age- and activity-matched control groups. This is a key feature when embarking on a critical review of fitness-intervention pedagogy, as there is something inherently efficacious about these interventions, despite the lack of reported pedagogical content.

### 4.1. Evidence of Pedagogy

The results of this review indicate minimal evidence of pedagogical reporting within the studies; however, it was not absent. Therefore, it is important to better understand the diversity of what pedagogical statements were included. Of those studies that did include practitioner behaviours, instruction was the most frequently used (*n* = 31). In those studies that stated ‘instruction’ was used, it was frequently the only defined practitioner behaviour identified. This establishes the behaviourist nature of the interventions, in that the practitioner told the participants what to do, how to do it, and when, in keeping with a reductionist, controlled research protocol. Based on the stated information within the methods, there were few circumstances where a two-way exchange took place between participant and intervention lead, with even less opportunity to exert any autonomy over what or how exercise was performed. The evidence derived from these studies reflects a narrow band of pedagogical practice, which is not reflective of effective practices [19,144]. In spite of limited pedagogical practice reported, the studies frequently reported positive effects on the fitness qualities trained.

Successfully delivering outcomes (increased physical fitness) only represents one component of the CPPRF [19]; the second element that this paper has considered is learner engagement. This concept is further enhanced through the consensus statement for physical literacy [145], which is that physical activity transcends more than simply movement, but includes and develops social, cognitive and affective elements. Consideration of participant engagement is absent in all but 1 study across the 106 reviewed. The assumptions made are that high-structure, dose–response type interventions do not achieve high engagement in younger children, and only increase physical fitness (outcomes). As may be expected, given the focus of these research studies, the breadth of outcome measures was heavily weighted towards physical outcomes and not towards the experience of the children undertaking them. The lack of engagement assessment is further evidence by the short-term, reductionist, behaviourist approach to these studies. Each study, individually, may not be criticised for this method of research, as their approaches were valid and rigorous, but the body of research as a collective may.

Williams et al. [137] did engage in qualitative interviews with their participants, undertaking different forms of warm-up protocols. In such an approach, the authors were able to form different judgements of the interventions, which is not possible with quantitative-only physiological data. More specifically, they found that participants expressed the opinion that they found a parkour-based warm-up (high in autonomy) was more fun to perform than their traditional warm-up. However, in doing so, Williams et al. [137] also offered further insights as to why there is a dearth of reporting for engagement and enjoyment. The specific constraints of conducting qualitative-data collection in primary-aged children is their ability to answer open questions in a valid way, due to their level of metacognition and language skills. This is coupled with the willingness of parents to provide consent for their child to be included in a research interview. Consequently, conducting primary qualitative research on the experiences of primary-aged children undertaking physical intervention studies is far more challenging than the relative simplicity of a fitness testing battery.

### 4.2. Motivational Climate

Faigenbaum et al. [13] recommended that fitness training, specifically resistance training, be conducted in an environment of skill mastery, exploration and fun, which has been deemed to be a “motivational climate” [146]. Analysis of the studies reviewed indicated that only 12 of the 106 had some form of statement that indicated a possible mastery climate, while the remaining 94 had no statements relating to motivational climate. The 106 studies did suggest that the “task-only”-based interventions were effective in improving the intended outcomes compared to control conditions. To add perspective on this, Goodway et al. [146] suggested that the constraints-based approach is most effective in developing children’s fitness, but not that other approaches were ineffective. By performing a specialised task that is designed to improve performance in aligned outcome measures, it would be anticipated that participants would improve. It is also probable that this rate of increase would be greater than their peers, who were engaged in a more generalised programme of activity. However, Goodway et al. [146] would argue that there is further improvement possible, over and above the adaptation seen from solely executing the task, if that task was situated in the appropriate psychological climate for that child.

If these studies are replicated as printed, then stifled psychological climates are created. To illustrate this, Ferrete et al. [81] applied a high-structured, 12-week intervention, with eight- and nine-year-old boys, using only modelling and instruction. Using this information alone, the climate created here was one of a behaviourist “teacher say, student do”, leaving little space for exploration, experimentation, imagination, or fun. From a hypothetical position, it is possible that these children may not develop as adaptable movers like those given a similar framework in a more constructivist, explorative and free psychological climate. Taking this point further, and considering the Sport England statement for physical literacy [145], yes, these participants moved, but it is questionable how much the environment afforded opportunities to develop thinking, feeling and connection. Furthermore, using the SDT [20], there is an absence of autonomy, and an intention to develop a sense of relatedness or competence, resulting in little change in intrinsic motivation to continue pursuing fitness beyond the study. Therefore, the research conducted by Ferrete et al. [81] and others like it, may at first sight be seen as positive, but, on reflection, it may now be viewed as an opportunity lost.

An alternative perspective could be taken that the control conditions, against which the study interventions were compared, were equally or further lacking in pedagogical planning, compared to the interventions. If such were true, then applying a clear focus, such as strength or speed, to a curriculum and activity structure, represents an advancement in the delivery of fitness development. Before arriving at such conclusions, consideration of the aims of the control conditions must be accounted for. This review is specifically exploring the development of fitness in children, yet in many cases the control conditions were the existing primary PE or sport curricula, which were much broader in their objectives. The principle of specificity explains that those children deliberately partaking in activities such as jump training would outperform children in jump measures, compared to those in general PE classes. In addition, the studies in this review were focused on the consequences of a highly specialised training intervention. They did not consider what control condition children improved on which the research group did not. When considering this question, all participant domains (psycho-motor, psycho-behavioural, psycho-social) should be considered, not just the intended physical fitness outcomes. For example, the control-condition children improved their object-control skills, motor creativity and social interactions, while the plyometric group increased their vertical jump. Equally, what skills declined in the intervention groups whilst they were focused on this single modality?

### 4.3. Translation and Implementation

The examination of pedagogy in children’s fitness is a timely and pertinent question to ask, given the decline in childhood fitness and the increased awareness of implementing evidence-based practice (EBP) [147]. The creation of activities for children’s fitness will emerge from this body of research and reflect what is reported, including any omissions or gaps. Using the RE-AIM Model [148] (reach, efficacy, adoption, implementation, maintenance) to view these studies, we must consider the ease and appropriateness with which practitioners may adopt and implement these interventions in their own contexts. Glasgow et al. [148] defined implementation as “… the extent to which a program is delivered as intended.” Practitioners can neither infer intentions nor assume certain pedagogies from the research evidence, but simply enact them as reported. Consequently, this review has highlighted significant gaps and omissions in this field of study, prohibiting the use of EBP within children’s fitness.

Using the applied model for research in sport sciences (ARMSS) (applied research model for sport sciences) [146], Bishop [149] stated research should include clear transparency of who delivered the intervention, how it was delivered, and the experiences of those within. Using the applied model for research in sport sciences (ARMSS) model of Bishop [149] stated that research should include clear transparency of who delivered the intervention, how it was delivered and the experiences of those within. The evidence within the 106 studies reviewed shows clearly that this way of reporting research has not been adopted. The reviewed studies show a consistent absence of how they were delivered as described by the three pedagogical variables (activity structure, practitioner behaviours, and motivational climate), relative to the needs of younger children. Where there was little evidence of how they were delivered, there was even less data relating to the experiences of the children within them. Therefore, the information available in the studies reviewed for this paper would not support the successful translation of the interventions, despite both their potential reach and efficacy.

Bishop [145] suggests that researchers should be cognizant of the range of considerations faced by practitioners when research is translated into contexts which are more “real world”. The research presented in this review would clearly fall short in this domain, and practitioners would see many barriers to the implementation of many of the interventions, based on the absence of pedagogical details reported. In many of studies reviewed, schoolteachers delivered these interventions, so it could be assumed that they would have implemented a pedagogy in keeping with their experiences, school culture, and context. However, in the absence of this being reported, consumers of this research have no way of knowing what this pedagogy may have been. Nor can they determine whether it was consistent between practitioners within the same study (e.g., between control and experimental groups). Furthermore, no assumptions can be made relating role to pedagogical practice; as stated by Randall [150], primary educators are under-prepared to deliver physical education, and therefore it cannot be assumed that pedagogical practices will be transferred into classes of different domains (i.e., cognitive and psychomotor). An EBP may only be formed around the evidence available and not what is missing yet assumed to be present.”

A final consideration when analysing this review, with specific reference to the translation and implementation, is the influence of the peer-review process for publication. All scientific publications have a specific focus, writing style and, potentially, word-count. Manuscripts are reviewed by editors, associate editors, and peers before they are accepted or rejected. Through this publication process, many amendments, additions and subtractions are made before finally becoming available to the consumer. The studies within this review are frequently published in journals which may place greater importance on the physiological, quantitative, “task, dose–response” data than the pedagogical and engagement data. In such articles it is customary to control variables and reduce measurement error, such that inferences can be made about the independent and dependent variables. In such a way, the inclusion of pedagogy muddies the waters of what may be a simpler research design. Consequently, the presented analysis of the studies may be less a reflection of the study design philosophy than an observation of the biases evident in the publication of such research. To address this point, the research community must accept the limitations of the current research approach and recognise that more contextually appropriate research design and subsequent reporting is essential.

### 4.4. Research Approach

The wider context of this paper is the development of fitness in children across the longer term (lifespan); however, this was not the purpose of those studies reviewed. The included papers were more short-term focused and explored the immediate impact of an intervention, lasting between four weeks and two years. Viewed through the SOLO taxonomy framework [151], it might appear that the authors considered their research problems as unistructural ones. Such an approach suggests that there is a single and simple solution to a problem. For example, to increase children’s jump height the solution needed is a plyometric training programme. This would represent a reductionist approach to studying childhood fitness development, through the controlling of non-intervention variables and determining a dose–response relationship. In this regard, it is perhaps understandable that pedagogy is less prevalent within the study methods reviewed, as it falls outside the unistructural perspective of measuring the impact of the exercise on the outcome.

There is nothing inherently incorrect in this reductionist approach, and in many circumstances it would be the correct research strategy. Controlling for pedagogical influence, through its minimisation, may be seen as advantageous, due to its potential impact on training performance. However, it does not serve practitioners trying to translate this evidence to the children they work with or replicate the study in different contexts. To expand on this point, the perspective of Goodway et al. [152] considers the development of fitness not as a unistructural one, as is evident in much of the reviewed research, but as multi-structural or relational. Goodway et al. [152] used the constraints-based model proposed by Newell et al. [153] to suggest the development of movement skills is a dynamic system that is influenced by the organism, task and environment. The environment includes multiple facets, including meteorological, physical and psychological. This constraints-based approach shows there is a complexity to the problem, and an interconnectivity between what is delivered, how it is delivered and the climate it is delivered in.

Within the research examined in this review, their purpose was to change children’s fitness through the execution of a simple and specific task (fitness training), but the environmental factors are typically limited to the physical space in which it was performed. The tasks themselves were again limited to the physical execution of movement(s). A more comprehensive way of reporting/approaching such research is to articulate tasks which also require the interaction between participants and practitioners, where they were able to make choices in how they interacted with the intervention. Furthermore, what was the psychological or motivational climate in which the participants undertook these fitness activities?

### 4.5. Limitations

The results of this review have shown that the published literature lacks pedagogical elements which may stimulate intrinsic motivation in children. However, the results of this review are limited to peer-reviewed, published fitness-intervention research, which constrains how far-reaching the implications of this review can be. The findings of this review cannot state that the declining fitness levels in children could or should be attributed to the pedagogical quality of all fitness training, nor that the interventions reviewed are representative of all training interventions worldwide. It is entirely probable that across the broad spectrum of fitness provision for 5–11-year-old children, there will be many instances of high quality and rich pedagogy. This raises the question: where did this pedagogy come from, if not the peer-reviewed literature in fitness interventions? This review has referred to EBP as an approach whereby research evidence forms one of three elements, supplemented by stakeholder perspectives and practitioner experiences. To address the constraints on the findings from this review, the other elements of EBP need exploration, in relation to the use of pedagogy within children’s fitness training.

### 4.6. Conclusion and Practical Applications

This paper aimed to review fitness-intervention studies used within children aged 5–11 years from a pedagogical perspective, using a systematic scoping review. Findings showed a broad range of fitness interventions were delivered within children, mostly demonstrating positive fitness outcomes. However, activity structures were predominantly highly structured, limiting opportunities for autonomy and exploration. Practitioner behaviours, essential for shaping learning and engagement, were omitted in 65% of studies and, when reported, mainly included instruction, reinforcing a behaviourist, one-way teaching model. Motivational climates, particularly those fostering mastery, autonomy, and relatedness, were referenced in only 12 studies. The interventions were found to be more akin to adult-appropriate training, and the lack of pedagogical reporting undermines the translational value of the research. It also risks creating fitness experiences misaligned with children’s developmental needs, potentially leading to negative associations with physical activity. Ultimately, effective child fitness interventions must move beyond what works, to embrace how it works. In embracing this change of emphasis, researchers and practitioners can ensure that physical improvements are matched by psychological growth and sustained participation.

To bridge this gap, future research must adopt relational and multi-structural approaches, recognizing fitness development as a dynamic interplay between task, environment, and learner. To evidence a shift in approach, research should explicitly integrate and report pedagogical strategies, aligning with frameworks such as the CPPRF [19] and the WHO–WHAT–HOW model [143]. Furthermore, manuscripts should follow the guidance of Bishop [149], and clarify who delivered the intervention (to whom), how this was delivered, and the experiences of those undertaking it. Finally, to complement the existing wealth of physical data, qualitative data should also be included, providing insights into children’s experiences and levels of engagement, enjoyment, and motivation.

## Figures and Tables

**Figure 1 sports-13-00309-f001:**
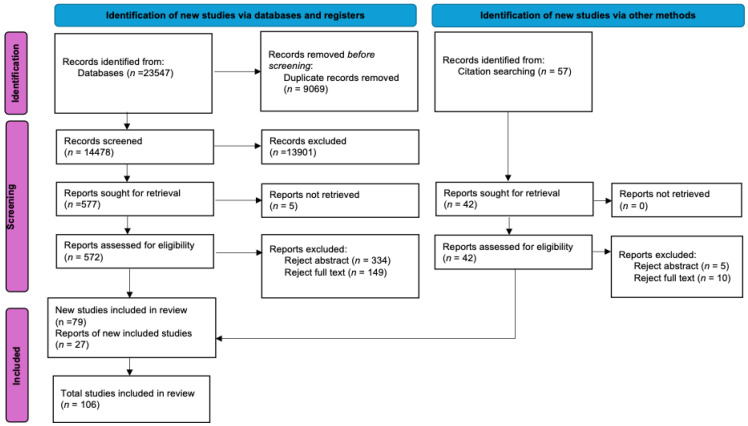
PRISMA flowchart of the systematic search strategy performed.

**Figure 2 sports-13-00309-f002:**
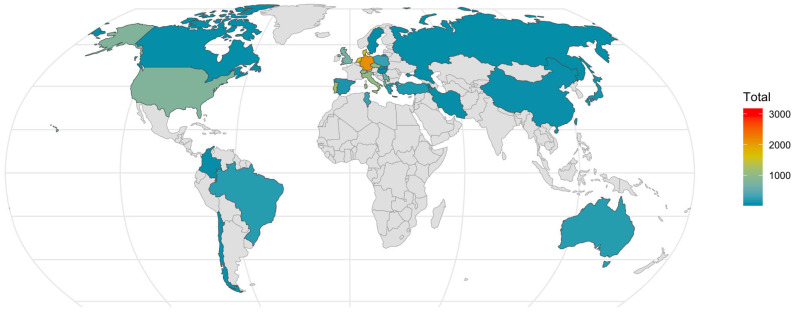
Global distribution of participants from the reviewed studies.

**Figure 3 sports-13-00309-f003:**
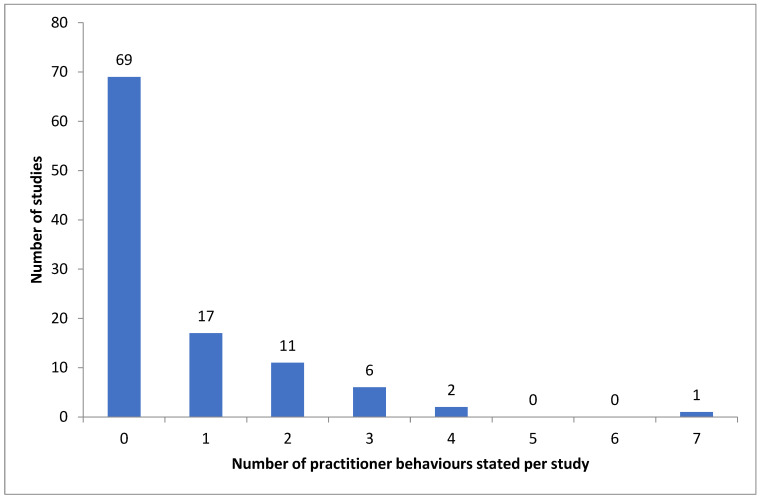
The frequency with which studies reported practitioner behaviours.

**Figure 4 sports-13-00309-f004:**
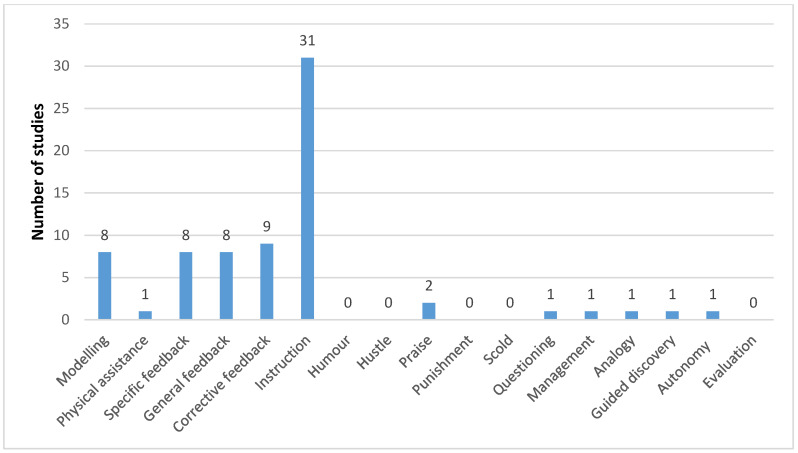
A histogram depicting the frequency of reported practitioner behaviours.

**Table 1 sports-13-00309-t001:** The inclusion and exclusion criteria applied to studies identified for review.

Inclusion Criteria	Exclusion Criteria
Peer-reviewed academic articles Primary research intervention studies Published between 2012 and 2023 Duration of intervention ≥ 4 weeks Children aged between 5 and 11 years. Published in English Children in mainstream education or sports participation	Grey literature, such as newspaper or magazine articles, blogs Published before 2012 Participants > 12 years (adolescents and adults) Systematic reviews, meta-analysis/regression, position stands, narrative reviews Specific clinical populations, such as those with an impairment (physical, sensory or learning impairment) medical condition such as, but not limited to, obesity, diabetes and cancer. Participants who are currently injured or recovering from injury

**Table 3 sports-13-00309-t003:** Analysis of stated practitioner behaviours.

	Modelling	Physical Assistance	Specific Feedback	General Feedback	Corrective Feedback	Instruction	Humour	Hustle	Praise	Punishment	Scold	Questioning	Management	Analogy	Guided Discovery	Autonomy	Evaluation	SUM
Abate Daga et al. [56]	0	0	0	0	0	0	0	0	0	0	0	0	0	0	0	0	0	0
Alberty and ČIllÍK [46]	0	0	0	0	0	0	0	0	0	0	0	0	0	0	0	0	0	0
Alesi et al. [57]	0	0	0	0	0	0	0	0	0	0	0	0	0	0	0	0	0	0
Almeida et al. [12]	0	0	0	0	0	0	0	0	0	0	0	0	0	0	0	0	0	0
Alonso-Aubin et al. [58]	0	0	0	0	0	0	0	0	0	0	0	0	0	0	0	0	0	0
Alves et al. [59]	0	0	0	0	0	0	0	0	0	0	0	0	0	0	0	0	0	0
Annesi et al. [60]	0	0	0	0	0	0	0	0	0	0	0	0	0	0	0	0	0	0
Arabatzi et al. [41]	0	0	0	0	0	0	0	0	0	0	0	0	0	0	0	0	0	0
Avetisyan et al. [61]	0	0	0	0	0	0	0	0	0	0	0	0	0	0	0	0	0	0
Barboza et al. [62]	0	0	0	0	0	0	0	0	0	0	0	0	0	0	0	0	0	0
Bogdanis et al. [63]	0	0	0	0	0	1	0	0	0	0	0	0	0	0	0	0	0	1
Boraczyński et al. [64]	0	0	0	0	0	0	0	0	0	0	0	0	0	0	0	0	0	0
Boraczyński et al. [65]	0	0	0	0	0	0	0	0	0	0	0	0	0	0	0	0	0	0
Bouguezzi et al. [66]	0	0	0	0	0	0	0	0	0	0	0	0	0	0	0	0	0	0
Bryant et al. [67]	1	0	0	0	0	1	0	0	0	0	0	0	0	0	0	0	0	2
Casolo et al. [68]	0	0	0	0	0	0	0	0	0	0	0	0	0	0	0	0	0	0
Cenizo-Benjumea et al. [69]	0	0	0	0	0	0	0	0	0	0	0	0	0	0	0	0	0	0
Chang et al. [70]	0	0	1	0	1	1	0	0	0	0	0	0	0	0	0	0	0	3
Chaouachi et al. [53]	0	0	0	0	0	0	0	0	0	0	0	0	0	0	0	0	0	0
Costa et al. [71]	0	0	0	0	0	0	0	0	0	0	0	0	0	0	0	0	0	0
Cunha et al. [55]	0	0	0	0	0	0	0	0	0	0	0	0	0	0	0	0	0	0
Cvejic and Ostojic [72]	0	0	0	0	0	0	0	0	0	0	0	0	0	0	0	0	0	0
De Greef et al. [47]	0	0	0	0	0	0	0	0	0	0	0	0	0	0	0	0	0	0
Donahoe-Fillmore and Grant [54]	0	1	0	0	0	1	0	0	0	0	0	0	0	0	0	0	0	2
Drouzas et al. [73]	0	0	0	0	0	0	0	0	0	0	0	0	0	0	0	0	0	0
Duncan et al. [74]	0	0	1	0	0	1	0	0	0	0	0	0	0	0	0	0	0	2
Duncan et al. [75]	0	0	1	0	0	1	0	0	0	0	0	0	0	0	0	0	0	2
Duncan et al. [76]	0	0	1	0	1	1	0	0	0	0	0	0	0	0	0	0	0	3
Eather et al. [77]	0	0	0	0	0	0	0	0	0	0	0	0	0	0	0	0	0	0
Elbe et al. [52]	0	0	0	0	0	1	0	0	0	0	0	0	0	0	0	0	0	1
Faigenbaum et al. [78]	1	0	0	0	1	1	0	0	0	0	0	0	0	0	0	1	0	4
Faigenbaum et al. [79]	0	0	0	0	0	0	0	0	0	0	0	0	0	0	0	0	0	0
Fernandes et al. [80]	0	0	0	0	0	0	0	0	0	0	0	0	0	0	0	0	0	0
Ferrete et al. [81]	1	0	0	0	0	1	0	0	0	0	0	0	0	0	0	0	0	2
Font-Lladó et al. [82]	0	0	1	1	1	0	0	0	0	0	0	0	0	0	0	0	0	3
Gallotta et al. [83]	0	0	0	0	0	0	0	0	0	0	0	0	0	0	0	0	0	0
Hammami et al. [84]	0	0	0	0	0	1	0	0	0	0	0	0	0	0	0	0	0	1
Hernández et al. [85]	0	0	0	0	0	1	0	0	0	0	0	0	0	0	0	0	0	1
Homeyer et al. [86]	0	0	0	0	0	0	0	0	0	0	0	0	0	0	0	0	0	0
Höner et al. [87]	0	0	0	1	0	1	0	0	0	0	0	0	0	0	0	0	0	2
Jaimes et al. [88]	0	0	0	0	0	0	0	0	0	0	0	0	0	0	0	0	0	0
Jarani et al. [89]	0	0	0	0	0	0	0	0	0	0	0	0	0	0	0	0	0	0
Keiner et al. [45]	0	0	0	0	0	0	0	0	0	0	0	0	0	0	0	0	0	0
Ketelhut et al. [90]	0	0	0	0	0	0	0	0	0	0	0	0	0	0	0	0	0	0
Koutsandréou et al. [91]	0	0	0	0	0	0	0	0	0	0	0	0	0	0	0	0	0	0
Larsen et al. [92]	0	0	0	0	0	0	0	0	0	0	0	0	0	0	0	0	0	0
Larsen et al. [93]	0	0	0	0	0	1	0	0	0	0	0	0	0	0	0	0	0	1
Latorre Román et al. [94]	0	0	0	1	0	0	0	0	0	0	0	0	0	0	0	0	0	1
Latorre Román et al. [95]	1	0	0	1	0	1	0	0	0	0	0	0	0	0	0	0	0	3
Lloyd et al. [42]	0	0	1	0	1	1	0	0	0	0	0	0	0	0	0	0	0	3
Lucertini et al. [96]	0	0	0	0	0	1	0	0	0	0	0	0	0	0	0	0	0	1
Marta et al. [97]	0	0	1	0	0	0	0	0	0	0	0	0	0	0	0	0	0	1
Marta et al. [98]	0	0	0	0	0	0	0	0	0	0	0	0	0	0	0	0	0	0
Marta et al. [99]	0	0	0	0	0	0	0	0	0	0	0	0	0	0	0	0	0	0
Marta et al. [100]	0	0	0	0	0	1	0	0	0	0	0	0	0	0	0	0	0	1
Marta et al. [101]	0	0	0	0	0	1	0	0	0	0	0	0	0	0	0	0	0	1
Marta et al. [102]	0	0	0	0	0	1	0	0	0	0	0	0	0	0	0	0	0	1
Martinez-Vizcaino et al. [103]	0	0	0	0	0	0	0	0	0	0	0	0	0	0	0	0	0	0
Marzouki et al. [104]	0	0	0	0	0	0	0	0	0	0	0	0	0	0	0	0	0	0
Mayorga-Vega et al. [105]	0	0	0	0	0	0	0	0	0	0	0	0	0	0	0	0	0	0
Menezes et al. [106]	0	0	0	0	1	1	0	0	0	0	0	0	0	0	0	0	0	2
Miller et al. [143]	0	0	0	0	0	0	0	0	0	0	0	0	0	0	0	0	0	0
MlChailidis et al. [107]	1	0	0	1	0	1	0	0	0	0	0	0	0	0	0	0	0	3
Moeskops et al. [108]	0	0	0	0	0	0	0	0	0	0	0	0	0	0	0	0	0	0
Moran et al. [109]	0	0	0	0	0	1	0	0	0	0	0	0	0	0	0	0	0	1
Ng et al. [110]	0	0	0	0	0	0	0	0	0	0	0	0	0	0	0	0	0	0
Orntoft et al. [111]	0	0	0	0	0	0	0	0	0	0	0	0	0	0	0	0	0	0
Parsons et al. [112]	1	0	0	0	0	1	0	0	0	0	0	0	0	0	0	0	0	2
Pinto-Escalona et al. [113]	0	0	0	0	0	0	0	0	0	0	0	0	0	0	0	0	0	0
Poveloy et al. [114]	0	0	0	0	0	0	0	0	0	0	0	0	0	0	0	0	0	0
Pomares-Nogueraet et al. [43]	0	0	0	0	0	0	0	0	0	0	0	0	0	0	0	0	0	0
Ramirez-Campillo et al. [115]	0	0	0	0	0	0	0	0	0	0	0	0	0	0	0	0	0	0
Redondo-Tebar et al. [38]	0	0	0	0	0	0	0	0	0	0	0	0	0	0	0	0	0	0
Reyes-Amigo et al. [117]	0	0	0	0	0	0	0	0	0	0	0	0	0	0	0	0	0	0
Richard et al. [116]	1	0	1	1	0	1	0	0	1	0	0	1	0	1	0	0	0	7
Rössler et al. [40]	0	0	0	0	0	0	0	0	0	0	0	0	0	0	0	0	0	0
Rössler et al. [118]	0	0	0	0	0	1	0	0	0	0	0	0	0	0	0	0	0	1
Sachetti et al. [48]	0	0	0	0	0	0	0	0	0	0	0	0	0	0	0	0	0	0
Sammoud et al. [119]	0	0	0	0	0	0	0	0	0	0	0	0	0	0	0	0	0	0
Savičević et al. [120]	0	0	0	0	0	0	0	0	0	0	0	0	0	0	0	0	0	0
Schlegel et al. [121]	0	0	0	0	0	0	0	0	0	0	0	0	0	0	0	0	0	0
Sijie et al. [39]	0	0	0	0	0	0	0	0	0	0	0	0	0	0	0	0	0	0
Skordal et al. [122]	0	0	0	0	0	0	0	0	0	0	0	0	0	0	0	0	0	0
St Laurent et al. [51]	0	0	0	1	0	0	0	0	0	0	0	0	0	0	0	0	0	1
Stupar et al. [123]	0	0	0	0	0	0	0	0	0	0	0	0	0	0	0	0	0	0
Tatsuo et al. [124]	0	0	0	0	0	0	0	0	0	0	0	0	0	0	0	0	0	0
Thompson et al. [125]	0	0	0	0	1	0	0	0	0	0	0	0	0	0	0	0	0	1
Tottori et al. [126]	0	0	0	0	0	0	0	0	0	0	0	0	0	0	0	0	0	0
Trajković and Bogataj [127]	0	0	0	0	0	1	0	0	0	0	0	0	0	0	0	0	0	1
Trajković et al. [128]	0	0	0	0	0	0	0	0	0	0	0	0	0	0	0	0	0	0
Trecroci et al. [129]	0	0	0	0	0	0	0	0	0	0	0	0	0	0	0	0	0	0
Tseng et al. [130]	1	0	0	0	0	0	0	0	0	0	0	0	0	0	0	0	0	1
Turgutet al, [131]	0	0	0	1	0	1	0	0	0	0	0	0	0	0	0	0	0	2
Vaczi et al. [132]	0	0	0	0	0	0	0	0	0	0	0	0	0	0	0	0	0	0
Vasileva et al. [133]	0	0	0	0	0	0	0	0	0	0	0	0	0	0	0	0	0	0
Vera-Assaoka et al. [134]	0	0	0	0	1	1	0	0	1	0	0	0	1	0	0	0	0	4
Wang et al. [135]	0	0	0	0	0	0	0	0	0	0	0	0	0	0	0	0	0	0
Waugh et al. [50]	0	0	0	0	0	0	0	0	0	0	0	0	0	0	0	0	0	0
Westblad et al. [136]	0	0	0	0	1	1	0	0	0	0	0	0	0	0	0	0	0	2
Williams et al. [137]	0	0	0	0	0	1	0	0	0	0	0	0	0	0	1	0	0	2
Yanci et al. [138]	0	0	0	0	0	0	0	0	0	0	0	0	0	0	0	0	0	0
Yanci et al. [44]	0	0	0	0	0	0	0	0	0	0	0	0	0	0	0	0	0	0
Yapıcı et al. [139]	0	0	0	0	0	0	0	0	0	0	0	0	0	0	0	0	0	0
Ye et al. [140]	0	0	0	0	0	0	0	0	0	0	0	0	0	0	0	0	0	0
Yildiz et al. [49]	0	0	0	0	0	0	0	0	0	0	0	0	0	0	0	0	0	0
Zarei et al. [141]	0	0	0	0	0	0	0	0	0	0	0	0	0	0	0	0	0	0
Zhang et al. [142]	0	0	0	0	0	0	0	0	0	0	0	0	0	0	0	0	0	0

**Table 4 sports-13-00309-t004:** Summary information of activity structure and motivational climate.

	Activity Structure	Format	Mastery	Autonomy	Relatedness	Climate
Abate Daga et al. [56]	High	Games (small sided)	No	No	No	Unclear
Alberty and ČIllÍK [46]	High	Mixed formats	No	No	No	Unclear
Alesi et al. [57]	Medium	Mixed formats	No	No	Yes	Mastery
Almeida et al. [12]	High	LEP	No	No	No	Unclear
Alonso-Aubin et al. [58]	High	Mixed formats	No	No	No	Unclear
Alves et al. [59]	High	LEP	No	No	No	Unclear
Annesi et al. [60]	High	Mixed formats	No	No	No	Unclear
Arabatzi et al. [41]	High	LEP	No	No	No	Unclear
Avetisyan et al. [61]	High	LEP	No	No	No	Unclear
Barboza et al. [62]	High	Mixed formats	No	No	No	Unclear
Bogdanis et al. [63]	High	Circuit training	No	No	No	Unclear
Boraczyński et al. [64]	High	Mixed formats	No	No	No	Unclear
Boraczyński et al. [65]	High	Mixed formats	No	No	No	Unclear
Bouguezzi [66]	High	LEP	No	No	No	Unclear
Bryant et al. [67]	High	Mixed formats	No	No	No	Unclear
Casolo et al. [68]	Medium	Games (small sided)	No	No	No	Unclear
Cenizo-Benjumea et al. [69]	Low	Games (pairs)	Yes	Yes	Yes	Mastery
Chang et al. [70]	High	LEP	No	No	No	Unclear
Chaouachi et al. [53]	High	LEP	No	No	No	Unclear
Costa et al. [71]	High	Mixed formats	No	No	No	Unclear
Cunha et al. [55]	High	LEP	No	No	Yes	Unclear
Cvejic and Ostojic [72]	High	Mixed formats	Yes	No	Yes	Unclear
De Greef et al. [47]	High	Interval training	No	No	No	Unclear
Donahoe-Fillmore and Grant [54]	High	LEP	No	No	No	Unclear
Drouzas et al. [73]	High	Not specified	No	No	No	Unclear
Duncan et al. [74]	High	LEP	No	No	No	Unclear
Duncan et al. [75]	High	LEP	No	No	No	Unclear
Duncan et al. [76]	High	Mixed formats	No	No	No	Unclear
Eather et al. [77]	Medium	Mixed formats	Yes	No	Yes	Mastery
Elbe et al. [52]	High	Mixed formats	No	No	No	Unclear
Faigenbaum et al. [78]	Medium	Circuit training	Yes	Yes	No	Mastery
Faigenbaum et al. [79]	High	Not specified	No	No	No	Mastery
Fernandes et al. [80]	High	Not specified	No	No	No	Unclear
Ferrete et al. [81]	High	LEP	No	No	No	Unclear
Font-Lladó et al. [82]	High	Mixed formats	No	No	No	Unclear
Gallotta et al. [83]	High	Mixed formats	Yes	No	No	Mastery
Hammami et al. [84]	High	LEP	No	No	No	Unclear
Hernández et al. [85]	High	LEP	No	No	No	Unclear
Homeyer et al. [86]	Medium	Games (individual)	No	No	No	Unclear
Höner et al. [87]	High	Mixed formats	Yes	No	No	Mastery
Jaimes et al. [88]	High	Mixed formats	No	No	No	Unclear
Jarani et al. [89]	High	Mixed formats	No	No	No	Unclear
Keiner et al. [45]	High	LEP	No	No	No	Unclear
Ketelhut et al. [90]	High	Mixed formats	No	No	No	Unclear
Koutsandréou et al. [91]	Medium	Games (individual)	No	No	No	Unclear
Larsen et al. [92]	High	Mixed formats	No	No	No	Unclear
Larsen et al. [93]	High	Mixed formats	No	No	No	Unclear
Latorre Román et al. [94]	High	Games (small sided)	No	No	No	Unclear
Latorre Román et al. [95]	High	LEP	No	No	No	Unclear
Lloyd et al. [42]	High	LEP	No	No	No	Unclear
Lucertini et al. [96]	High	Not specified	No	No	No	Unclear
Marta et al. [97]	High	LEP	No	No	No	Unclear
Marta et al. [98]	High	LEP	No	No	No	Unclear
Marta et al. [99]	High	Mixed formats	No	No	No	Unclear
Marta et al. [100]	High	LEP	No	No	No	Unclear
Marta et al. [101]	High	LEP	No	No	No	Unclear
Marta et al. [102]	High	LEP	No	No	No	Unclear
Martinez-Vizcaino et al. [103]	High	Games (small sided)	No	No	No	Unclear
Marzouki et al. [104]	High	LEP	No	No	No	Unclear
Mayorga-Vega et al. [105]	High	Circuit training	No	No	No	Unclear
Menezes et al. [106]	High	LEP	No	No	No	Mastery
MlChailidis et al. [107]	High	LEP	No	No	No	Unclear
Moeskops et al. [108]	High	LEP	No	No	No	Unclear
Moran et al. [109]	High	LEP	No	No	No	Unclear
Ng et al. [110]	High	LEP	No	No	No	Unclear
Orntoft et al. [111]	High	Mixed formats	No	No	No	Unclear
Parsons et al. [112]	High	Not specified	No	No	No	Unclear
Pinto-Escalona et al. [113]	High	Mixed formats	No	No	No	Unclear
Poveloy et al. [114]	High	Mixed formats	No	No	No	Unclear
Pomares-Nogueraet et al. [43]	High	Not specified	No	No	No	Unclear
Ramirez-Campillo et al. [115]	High	LEP	No	No	No	Unclear
Redondo-Tebar et al. [38]	High	Games (small sided)	No	No	No	Unclear
Reyes-Amigo et al. [117]	High	Mixed formats	No	No	No	Unclear
Richard et al. [116]	Medium	Mixed formats	Yes	Yes	No	Mastery
Rössler et al. [40]	High	Mixed formats	No	No	No	Unclear
Rössler et al. [118]	High	Mixed formats	No	No	No	Unclear
Sachetti et al. [48]	High	Mixed formats	No	No	No	Unclear
Sammoud et al. [119]	High	LEP	No	No	No	Unclear
Savičević et al. [120]	Medium	Mixed formats	Yes	Yes	Yes	Mastery
Schlegel et al. [121]	High	resistance training	No	No	No	Unclear
Sijie et al. [39]	High	Interval training	No	No	No	Unclear
Skordal et al. [122]	High	Games (small sided)	No	No	No	Unclear
St Laurent et al. [51]	High	LEP	No	No	No	Unclear
Stupar et al. [123]	High	Not specified	No	No	No	Unclear
Tatsuo et al. [124]	Medium	Games (individual)	No	No	No	Unclear
Thompson et al. [125]	High	Not specified	No	No	No	Unclear
Tottori et al. [126]	High	Interval training	No	No	No	Unclear
Trajković and Bogataj [127]	High	Not specified	No	No	No	Unclear
Trajković et al. [128]	High	Not specified	No	No	No	Unclear
Trecroci et al. [129]	High	LEP	No	No	No	Unclear
Tseng et al. [130]	High	Not specified	No	No	No	Unclear
Turgutet al, [131]	High	LEP	No	No	No	Unclear
Vaczi et al. [132]	High	LEP	No	No	No	Unclear
Vasileva et al. [133]	High	Circuit training	Yes	No	Yes	Unclear
Vera-Assaoka et al. [134]	High	LEP	No	No	No	Unclear
Wang et al. [135]	High	Mixed formats	No	No	No	Unclear
Waugh et al. [50]	High	LEP	No	No	No	Unclear
Westblad et al. [136]	High	LEP	No	No	No	Unclear
Williams et al. [137]	Medium	Mixed formats	Yes	Yes	No	Mastery
Yanci et al. [138]	High	Mixed formats	No	No	No	Unclear
Yanci et al. [44]	High	Mixed formats	No	No	No	Unclear
Yapıcı et al. [139]	High	Not specified	No	No	No	Unclear
Ye et al. [140]	Medium	Exergaming	No	Yes	No	Unclear
Yildiz et al. [49]	High	LEP	No	No	No	Unclear
Zarei et al. [141]	High	Not specified	No	No	No	Unclear
Zhang et al. [142]	High	Mixed formats	No	No	No	Unclear

LEP = Linear exercise progression.

## Data Availability

Not applicable.

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
