# Peer review of "Developing the Physical Fitness of Children: A Systematic Scoping Review of Pedagogy in Research"

_sports, 2025, doi:10.3390/sports13090309_

Round 1
Reviewer 1 Report
Comments and Suggestions for Authors
Dear Editor,
Thank you for the opportunity to review this paper. The manuscript addresses a relevant and timely topic: the decline in fitness levels in children, despite the many known benefits of physical activity. The originality of the paper lies in the analysis of the pedagogical dimension of interventions, an element often neglected in literature. The message is very important: more attention to pedagogy is needed to turn these protocols into effective interventions also in the long term and in the real world. However, I would like to provide some suggestions to improve the quality of your manuscript:
Introduction-
The introduction is unclear. Several concepts have not been explained which makes the interpretation of manuscript difficult for non-experts.
You should better specify the connection problem - objective - research hypothesis.
It should be justified why the methodological choice of scoping review rather than other types such as systematic etc. should be chosen.
Some expressions are redundant and grammatically incorrect.
Method
The Methods section is extensive, well structured and adhering to the PRISMA guidelines, giving methodological soundness to the work.
Results
Review the tables because sometimes there are repetitions Ex. Almeida et. al [52]
Caption on figure 4 goes after figure 3. Each figure should be preceded by description. Ditto for the tables. Since the tables are very large it is difficult to understand the results if descriptions are put randomly.
Discussions
Discussions are long and repetitive.
I would suggest breaking it down into subsections and by concepts and conclude it with
a summary of the key concept. There are grammatical errors such as “findings off this review.”
Conclusions
Reframe the conclusions better by emphasizing the main findings through a synthesis.
Author Response
Dear reviewer,
Thank you for taking your time and effort in the review of this paper, and the consider recommendations you have made. We welcome all comments that will improve the quality of the paper and the impact it may have. We have included a summary which outlines those specific actions you have suggested alongside our responses and locations within the paper where they have been taken.
We have paid particular attention to the introduction to further clarify concepts which may not familiar to some readers and enhance the connection between the research problem and study aims. However, as a scoping review, we were not hypothesizing there would be a specific outcome, nor were we testing a hypothesis. We have also justified the use of a scoping review.
|
Reviewer comment |
Response (inc lines & Action) |
|
R1. Sevral concepts have not been explained which makes the interpretation of manuscript difficult for non-experts. |
Changes made to the introduction to more clearly introduce several concepts |
|
R1. You should better specify the connection problem - objective - research hypothesis |
We have made edits to the intro to may clearly state the problem and align to the study aim. However, due to the nature of the scoping review used, it is common to report a hypothesis |
|
R1. It should be justified why the methodological choice of scoping review rather than other types such as systematic etc. should be chosen. |
Lines 68-78.
A statement has been added to state why a scoping review was more appropriate than a systematic review. |
|
R1. Some expressions are redundant and grammatically incorrect. |
Changes made throughout |
|
R1. Review the tables because sometimes there are repetitions Ex. Almeida et. al [52] |
Duplicated paper references removed. |
|
R1. Caption on figure 4 goes after figure 3. |
In preparing the manuscript we adhered to the APA guidelines and have inserted captions below all figures three and four proceed the figures, as per academic convention. These were subsequently formatted by Sports into the desired manuscript template before being sent out for review. |
|
R1. Each figure should be preceded by description. Ditto for the tables. |
We have indicated below where each of the tables and figures were described within the text. In each case these comments precede either the table or figure to which they refer and provide, we feel, sufficient information about what is presented.
Lines 126-129 Search results were excluded where the participants were not within the specified age ranges, which are presented in table 1, for example interventions conducted in pre-schools or high schools. The interventions were sport specific or
Lines 226-227 Following the application of the inclusion and exclusion criteria, 106 studies were included for review as illustrated in Figure 1.
Lines 237-238 Table 2 presents a summary of the studies including participants, intervention and outcome measures. Across all 106 studies, 18,321 children were included within the review. These were from 30 countries, spanning six continents and the global distribution of these participants can be seen in Figure 2
Lines 305-315 Figure 3 presents the frequency distribution for the number of coach behaviours stated in each of the studies. The highest frequency of behaviours reported within a study method was seven (111), which was a moderately structured intervention with 9 year old children. Figure 3 is a histogram that illustrates the frequency with which intervention research reports prescribed coach behaviours. This image indicates a clear skew in the data towards the implementation of minimal coach behaviours. From the 34 studies that reported coach behaviours, instruction was the most frequently deployed (n=30) followed by corrective feedback, illustrated in Figure 4. |
|
R1. Since the tables are very large it is difficult to understand the results if descriptions are put randomly. |
Please see the point above, each table and figure have captions located according to APA guidelines and they are referred to within the text.
We cannot identify any instances where the descriptions or captions are randomly placed within the manuscript. |
|
R1. Discussions are long and repetitive. I would suggest breaking it down into subsections and by concepts and conclude it with |
Significant changes have been made to the discussion to reduce the content and by clear on the findings and applications of the work
Several sub-headings have been inserted and the sections have been amended to improve their readability.
Additionally, following comments from other reviewers, the discussion and conclusions have been updated to be more focused and concise. |
|
R1. There are grammatical errors such as “findings off this review.” |
Grammatical, syntax and spelling errors have been corrected. |
Reviewer 2 Report
Comments and Suggestions for Authors
A highly useful paper for all authors who will encounter this issue in the future. I believe that the title should be reformulated and that the term coach's behaviour should also be found in it, because a lot of attention is paid to it in the work.
Below I have a few minor suggestions for correction:
- Match the font of text and graphics
- On chart 4, give the y-axis title
- In Table 3, give the column headings on each page. This type of behaviour cannot be tracked
- Same for Table 4.
Author Response
Dear reviewer,
Thank you for taking your time and effort in the review of this paper, and the consider recommendations you have made. We welcome all comments that will improve the quality of the paper and the impact it may have. We have included a summary which outlines those specific actions you have suggested alongside our responses and locations within the paper where they have been taken.
Your comments were valuable and the manner in which the information is presented is vital to ensure the readers can extract the key points easily and fully. However, in some cases we beholden to the manner in which the publishing journal formats the manuscript to meet their templates.
|
Reviewer comment |
Response (inc lines & Action |
|
R2. Match the font of text and graphics |
All fonts were amended in the submitted file |
|
R2. On chart 4, give the y-axis title |
Axis label added |
|
R2. In Table 3, give the column headings on each page. Same for Table 4. |
The configuration of tables is at the discretion of the publisher. A review of recent review articles, including a similar range of studies did not amend tables to include headings for each page. |
|
R2. This type of behaviour cannot be tracked |
Can the reviewer please provide more information on this point? We are confused as to whether this is about how difficult it is to track individual behaviours in table 3 without additional headings, or if there is a conceptual point about what behaviours are measurable.
If it is the latter (acknowledging we have addressed column headers in the previous comment), then all the behaviours selected are from previously developed behavioural frameworks and are observable. |
Reviewer 3 Report
Comments and Suggestions for Authors
Thank you for the opportunity to review this very interesting manuscript. Below are my suggestions which hopefully will be useful to the authors in the redrafting phase.
Abstract
The suggestion here is to remove sub-headings, as the readership will be able to discern the key elements of the paper from the way in which the constituent statements have been written.
Introduction
This section has been well written with a clear focus on the topic at hand.
Method
The method has been outlined and justified nicely.
Methodology
Within this section it is noted that the systematic scoping review was conducted in alignment with the PRISMA guidelines. Within these guidelines, the literature search was undertaken between the lead author and an independent reviewer. However, it would be helpful to include the results of these two reviewers’ analyses of manuscripts; namely, the inter-rater reliability agreement metric (or some kind of result indicating the extent to which these two people agreed/disagreed with the inclusion/exclusion of papers in the scoping review). Without this layer of transparent detail, the authorship is left to question the extent to which any potential disagreements of manuscript inclusion/exclusion were reconciled.
Results
The results have been presented clearly and coherently.
Discussion
(i) This section suffers from many issues with syntax, expression, spelling, and omitted words which detracts from the quality of the paper and hampers the readability overall.
(ii) The aim of the paper/review is mentioned four times in this section, which is unnecessary repetition. If the aim must be mentioned several times, then the suggestion is to ensure that it reads consistently and not as though there are several aims. Conversely, and if there are multiple aims of the paper/review, then the authors will need to adjust the first mention of the aims in the front matter of the paper.
Conclusions and Practical Applications
This section ties together the key points of the paper nicely.
References
The references are relevant to this work and have been cited appropriately.
Author Response
Dear reviewer,
Thank you for taking your time and effort in the review of this paper, and the consider recommendations you have made. We welcome all comments that will improve the quality of the paper and the impact it may have. We have included a summary which outlines those specific actions you have suggested alongside our responses and locations within the paper where they have been taken.
We have paid particular attention to the discussion section to ensure the findings are discussed effectively in relation to the sole aim of the study.
|
Reviewer comment |
Response (inc lines & Action |
|
The suggestion here is to remove sub-headings, as the readership will be able to discern the key elements of the paper from the way in which the constituent statements have been written. |
Lines 3-18 The abstract has been reformatted to remove the sub-headings |
|
Within this section it is noted that the systematic scoping review was conducted in alignment with the PRISMA guidelines. Within these guidelines, the literature search was undertaken between the lead author and an independent reviewer. However, it would be helpful to include the results of these two reviewers’ analyses of manuscripts; namely, the inter-rater reliability agreement metric (or some kind of result indicating the extent to which these two people agreed/disagreed with the inclusion/exclusion of papers in the scoping review). Without this layer of transparent detail, the authorship is left to question the extent to which any potential disagreements of manuscript inclusion/exclusion were reconciled. |
Lines 100 – 108. This section of the methods has been amended to report the agreement levels between reviewers. |
|
This section suffers from many issues with syntax, expression, spelling, and omitted words which detracts from the quality of the paper and hampers the readability overall. |
The discussion has been re-written to improve the syntax, expression, spelling, and omitted words |